# Replication of ecologically relevant hydrological indicators following a modified covariance approach to hydrological model parameterisation

Annie Visser-Quinn[1], Lindsay Beevers[1], Sandhya Patidar[1]

[1]Institute for Infrastructure and Environment, Heriot-Watt University, Edinburgh, EH14 4AS, UK

*Correspondence to*: Annie Visser-Quinn (a.visser-quinn@hw.ac.uk)

**Abstract.** Hydrological models can be used to assess the impact of hydrologic alteration on the river ecosystem. However, there are considerable limitations and uncertainties associated with the replication of ecologically relevant hydrological indicators. Vogel and Sankarasubramanian's 2003 (Water Resources Research) covariance approach to model evaluation and parameterisation represents a shift away from algorithmic model calibration with traditional performance measures (objective functions). Using the covariance structures of the observed input and simulated output time-series, it is possible to assess whether the selected hydrological model is able to capture the relevant underlying processes. From this plausible parameter space, the region of parameter space which best captures (replicates) the characteristics of a hydrological indicator, may be identified. In this study, a modified covariance approach is applied to five hydrologically diverse case study catchments with a view to replicating a suite of ecologically relevant hydrological indicators identified through catchment-specific hydroecological models. The identification of the plausible parameter space (here n ≈ 20) is based on the statistical importance of these indicators. Evaluation is with respect to performance and consistency across each catchment, parameter set, and the 40 ecologically relevant hydrological indicators considered. Timing and rate of change indicators are the best and worst replicated respectively. Relative to previous studies, an overall improvement in consistency is observed. This study represents an important advancement towards the robust application of hydrological models for ecological flow studies.

## 1 Introduction

Increases in societal water demand and climatic variability raise questions over the long-term sustainability of water resources (Gleick, 1998; Klaar et al., 2014; Davis et al., 2015; Gleick, 2016). As the ecological role of flow is better understood, it has become widely acknowledged as the major determinant of the ecological health of the riverine ecosystem (e.g. Power et al. (1995); Lytle and Poff (2004); Arthington et al. (2006)). Consequently, changes to flow threatens both the ecological health of rivers and their ability to provide the vital ecosystem services upon which humans depend (Vörösmarty et al., 2010; Arthington, 2012).

Beginning in the late 1940s in the United States, the need to balance the conflicting demands of both human society and those of the ecosystem saw the emergence of the environmental flow movement. Environmental flows have been defined under the

Brisbane Declaration (2007) as: *"...the quantity, timing, and quality of water flows required to sustain freshwater and estuarine ecosystems and the human livelihood and well-being that depend on…"*. Tharme (2003) documented that over 200 formal environmental flow assessment methods had been developed.

Quantifying the relationship between flow and ecology is pivotal for the determination of environmental flows (Bunn and Arthington, 2002; Arthington et al., 2006; Poff et al., 2010; McManamay et al., 2013). Richter et al. (1996) identified five facets of the flow regime required to support the riverine ecosystem: magnitude, frequency, duration, timing and rate of change. Alteration of the flow regime invariably leads to significant ecologic change. To date, over 200 ecologically relevant hydrologic indices (ER HIs) have been proposed (Olden and Poff, 2003; Monk et al., 2006; Thompson et al., 2013). Poff et al. (2010) and Peters et al. (2012) each describe environmental flow frameworks, which call for the determination of ER HIs via hydrological model simulations of flow. At the time of publication (of these frameworks), the application of hydrological models for the determination of ER HIs was in its infancy (Knight et al., 2011). Indeed, early work was largely based on regional statistical approaches which had been in use since the 1960s in the United States (for the determination of water resource relevant HIs; for example, see Knight et al. (2011) and Carlisle et al. (2010)). Murphy et al. (2012) compared such ER HIs against those determined from simulated flows, finding that, without targeted calibration to specific HIs, *"the widespread application of general hydrologic models to ecological flow studies is problematic"* (p. 667). However, such statistical approaches are unsuitable when assessing the impact of hydrological change on the river ecosystem (e.g. as a result of engineering intervention or under a changed climate) or for the simulation of ecological flows in ungauged catchments. A hydrological modelling approach is thus necessary.

Model performance and consistency are watchwords for this study. After Euser et al. (2013), model performance is defined as the ability to mimic the behaviour of catchment hydrological processes; consistency represents the ability of the hydrological model to reproduce a suite of ER HIs across parameter sets, hydrological models and catchments.

Significant bias has been observed in hydrological models calibrated following algorithmic model calibration with objective functions and performance measures (Grayson and Blöschl (2001); Blöschl and Montanari (2010); Westerberg et al. (2011); Pushpalatha et al. (2012)); hereafter this is termed the 'traditional approach'. For example, when evaluating the suitability of model simulated HIs (six water resource relevant HIs and 32 ER HIs), Shrestha et al. (2014) observed that water resource relevant HIs were well-replicated whilst notable differences were observed for ER HIs related to the facets of the flow regime duration and rate of change. Informed by recent advances in hydrological modelling more generally (Seibert, 2000; Efstratiadis and Koutsoyiannis, 2010), Vis et al. (2015) compared the ability of single and multi-criteria objective functions to replicate twelve ER HIs. The best performance was achieved with multi-criteria objective functions, though a consistent negative bias was observed. Despite these advances, overall performance was inconsistent, being dependent upon the ER HI considered. Blöschl and Montanari (2010) observed that the reliability of hydrological modelling approaches which try to 'model everything' is analogous to simply 'throwing the dice'. To address this, they call for a move towards simpler models, tuned to focus on specific characteristics of the flow regime; successful applications of such an approach include Westerberg et al. (2011). Most recently, Pool et al. (2017) considered an array of multi-criteria objective functions using Nash Sutcliffe

Efficiency (NSE) and 13 ER HIs. Results were positive, with ER HIs generally well-replicated, though the transposability of the model was subject to greater variability. Those ER HIs not explicitly included in the objective function exhibited greatest uncertainty overall.

The past ten years has seen the replication of ER HIs evolve from statistical approaches to single and multi-objective rainfall-runoff modelling. Whilst improvements have been notable, to date no approach has been able to achieve performance and consistency concurrently, raising questions as to whether these approaches are able to achieve the 'right answer for the right reasons'. Pool et al. (2017) highlight two points which remain unaddressed: (1) a need to determine which ER HIs are relevant in order to guide model parameterisation; and (2) laborious recalibration of the hydrological model is necessary if the suite of HIs is changed. In addition, model evaluation in these studies is singularly focussed on the goodness of fit of the observed-simulated data, while the ability of the hydrological model to capture the relevant hydrological processes is not considered. In this paper we look to redress these limiting factors through the application of a modified covariance approach. The objective of Vogel and Sankarasubramanian's (2003) covariance approach is to identify the plausible parameter space which captures (replicates) the characteristics of a specified HI. This is achieved by focussing on the ability of the hydrological model to capture the observed covariance structure of the input and output time-series. The use of covariance relationships in this way is not new, with examples including the modelling of ice sheets (Wu et al., 2010) and ocean salinity (Haines et al., 2006). Vogel and Sankarasubramanian's covariance approach is limited by its focus on a single HI, preventing its use for the determination of a suite of ER HIs. This paper builds on the covariance approach, adapting the methodology to consider a suite of ecologically relevant hydrological indicators; the determination of these ER HIs is based on the outcomes of hydroecological modelling using an Information Theory approach. To determine the ability of the modified covariance approach in replicating ER HIs, the method is applied to five case study catchments across the UK using the daily models from the GR (Génie Rural) suite of hydrological models (GR4J, GR5J and GR6J, 4-6 free parameters; Coron et al. (2018)).

## 2 Methods

### 2.1 Study areas

The UK is home to a wide range of hydrological environments, with 18 different river types (based on catchment area, mean altitude and geology) specified under the Water Framework Directive (Rivers Task Team, 2004). Therefore, to illustrate the generality of the modified covariance approach, it is necessary to apply the proposed methodological approach to a range of catchments with differing characteristics (Andreassian et al., 2006; Gupta et al., 2014). Hydroecological models inform the parameterisation of the hydrological models. A mismatch between the co-location of sampling sites as well as the length of time-series is a known limiting factor in hydroecological modelling (Monk et al., 2006; Knight et al., 2008). In the UK, this may be addressed, in part, by the recent publication of the UK BIOSYS archive (long-term ecological monitoring data from across England and Wales; Environment Agency (2018)). In this study, ecological and flow time-series were paired, and catchments assessed in terms of length of the paired dataset (> 10 years), number of sampling sites (> 5), location, catchment

area, altitude, catchment steepness (m/km), baseflow index (BFI) and land use. A total of five catchments were selected across the UK, from the north of Scotland to the south-west of England (Fig. A1); catchment characteristics are summarised in Table 1.

**Table 1. Summary of case study catchment characteristics. Catchment steepness is unavailable for the Tarland Burn.**

| | | Tarland Burn | River Trent | River Ribble | River Nar | River Thrushel |
|---|---|---|---|---|---|---|
| *Flow gauge and catchment* | *Location* | Aboyne | Stoke-On-Trent | Arnford | Marham | Hayne Bridge |
| | *Longitude* | -2.7758 | -2.1624 | -2.2471 | 0.5472 | -4.2424 |
| | *Latitude* | 57.0777 | 53.0175 | 53.9962 | 52.6783 | 50.6584 |
| | *Altitude, gauge (mAOD)* | 125 | 113 | 117 | 5 | 67 |
| | *Altitude, max (mAOD)* | 616 | 331 | 691 | 85 | 273 |
| | *Catchment steepness (m/km)* | - | 68 | 100 | 23 | 94 |
| | *Bedrock geology* | Mafic and felsic igneous | Mud/siltstone, sandstone | Mud/siltstone, sandstone; limestone | Chalk | Mud/siltstone, sandstone |
| | *Baseflow index* | 0.66 | 0.44 | 0.25 | 0.91 | 0.39 |
| | *Drainage area (km²)* | 70.9 | 53.2 | 204 | 153 | 57.6 |
| | *Principal land use* | Mountain, heath and bog | Urban and grassland | Grassland | Arable and horticulture | Grassland |
| *Data* | *Years* | 2003-2016 | 1989-2016 | 2000-2016 | 1961-2015 | 1989-2016 |
| | *Flow data source* | JHI (2018) | NRFA (2018) | | | |
| | *Climate data source* | | Met Office (2018a) and Met Office (2018b) | | | |

## 2.2 Hydrological model

The principle of parsimony, known as Occam's razor, posits that a solution should be no more complex than necessary. In the context of hydrological modelling, model simplicity relative to performance is thus made key (Kokkonen and Jakeman, 2002; Perrin et al., 2003; Beven, 2012). To this end, the three lumped models from the GR-J series of daily hydrological models was selected (Perrin et al., 2003): GR4J, GR5J and GR6J (4, 5 and 6 free parameters respectively; Perrin et al. (2003); Le Moine (2008); Pushpalatha et al. (2011)). The GR-J series of models have been applied in a variety of hydrological contexts, including climate change impact assessment, water resources forecasting and prediction in ungauged catchments; for examples, see: Rojas-Serna et al. (2006); Perrin et al. (2008); Coron et al. (2012); Smith et al. (2012); Coron et al. (2017).

The three models are based on soil moisture accounting (Fig. A2); precipitation and potential evapotranspiration serve as input. Water is directed to a production store with capacity $x1$ mm, split into routed and direct components, and input to unit hydrographs with time base $F(x4)$ days. The routed flow is directed to a routing store with capacity $x3$ mm. Finally, a groundwater exchange term $F(x2)$, acts on the routed and direct flow components. The total flow, $Q$, is the sum of the routed and direct flow. To improve general model efficiency (Anderson Michael et al., 2004; Hughes, 2004), GR5J sees the addition

of the inter-catchment exchange threshold, $x5$, a function representing the interaction between channel and aquifer flows (Le Moine, 2008). To improve simulations of low flows, the GR6J model includes a parallel store with capacity $x6$ mm (Pushpalatha et al., 2011). The models are applied using the R package *airGR* (Version 1.0.15.2; Coron et al. (2017); Coron et al. (2018). Parameter limits are summarised in Table A1.

## 2.3 Determination of ecologically relevant hydrological indicators

The ER HIs were determined based on the outcomes of hydroecological modelling for each catchment. Following Visser et al. (2018), hydroecological models were developed using multiple linear regression with an information theory (IT) approach; see Appendix A.2 for details. The IT approach provides a measure of the statistical importance of each ER HI. Consequently, more conclusive statements may be made with regards to the model and the relevance of the ER HIs. To reflect seasonality in the flow regime, the indices are differentiated by hydrological season: winter (ONDJFM) and summer (AMJJAS). Definitions of the ER HIs included in the hydroecological models, and their importance, are available in Table B1. A summary of the distribution of the ER HIs per facet of the flow regime, season and river is provided in Table 2.

Table 2. Number of ER HIs per facet of the flow regime, season (W and S denote summer and winter respectively) and river. Sum totals are detailed in the final columns and rows.

| Facet of the flow regime | | Tarland Burn W | Tarland Burn S | River Ribble W | River Ribble S | River Trent W | River Trent S | River Nar W | River Nar S | River Thrushel W | River Thrushel S | Sum per facet |
|---|---|---|---|---|---|---|---|---|---|---|---|---|
| (M) Magnitude | *Statistic* | 1 | 1 | 1 | 2 | 1 | | | 1 | 2 | | 9 |
| | *Ratios – Log quantile* | | | | | 2 | 1 | | | | 1 | 4 |
| | *Ratios – Median-quantile* | | | | 4 | 2 | | | 3 | 1 | 2 | 12 |
| | *Monthly* | 2 | | | | 1 | | | | 1 | | 4 |
| (D) Duration | | | 2 | 1 | | 2 | | | | 1 | | 6 |
| (F) Frequency | | 1 | | 1 | 1 | 1 | 1 | | | 2 | | 7 |
| (T) Timing | | | 1 | | 2 | 2 | | | | 1 | | 6 |
| (R) Rate of change | | | | | 1 | | 1 | 1 | 1 | 1 | | 5 |
| *Sum per season per river* | | 4 | 4 | 3 | 10 | 9 | 4 | 2 | 5 | 9 | 3 | 53 |

## 2.4 Covariance approach

Continuous (daily) time-series of mean flow, precipitation and potential evapotranspiration serve as input to the hydrological models; flow and climate data availability are summarised in Table 1 previously. Potential evapotranspiration was estimated using a temperature-based PE model (Oudin et al., 2005).

The covariance approach was developed by Vogel and Sankarasubramanian (2003), where the aim was to replicate a specific HI rather than the flow time-series. The modification of the covariance approach in this study allows for the consideration of a suite of ecologically relevant HIs. The modified covariance approach is implemented over three stages (Fig. 1); stages 1 and

2 are as in Vogel and Sankarasubramanian (2003), with the exception that multiple ER HIs are calculated, with the final stage representing the modification introduced in this study.

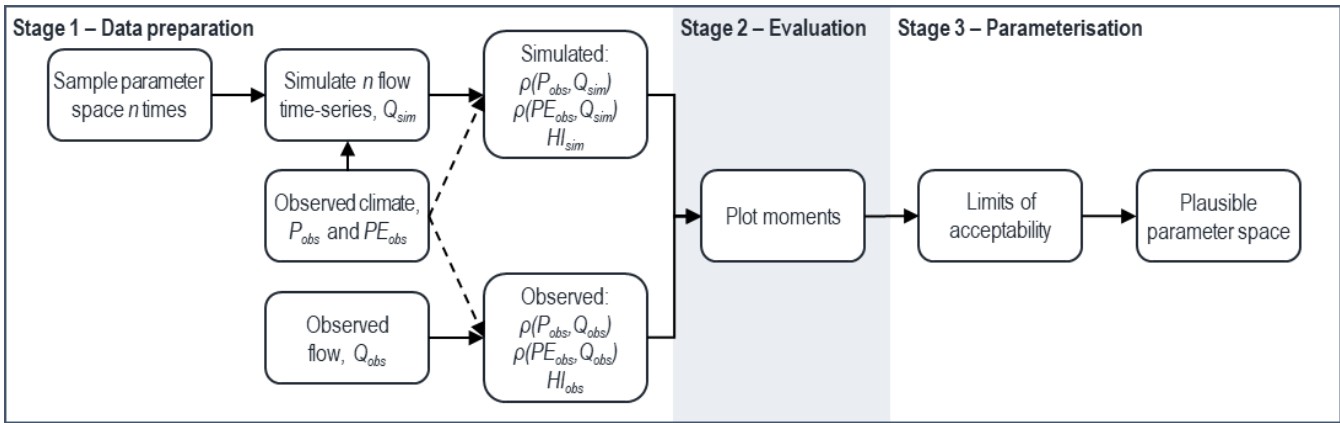

**Figure 1. Overview of the three stages of the modified covariance approach to model parameterisation.**

*Stage 1, data preparation:* The parameter space of the three hydrological model structures was sampled within the limits specified in Table A1. With a view to addressing both parameter sensitivity (Tong and Graziani, 2008; Wu et al., 2017) and the number of parameter sets considered, the parameter space was sampled uniformly based on Sobol quasi-random sequences (a Quasi-Monte Carlo method). The River Nar catchment served as the 'proof-of-concept', consequently, for this catchment, 100,000, 150,000 and 200,000 independent parameter sets were selected for the GR4J, GR5J and GR6J hydrological models
respectively; for the remaining four catchments, 10,000 parameter sets were considered (per hydrological model).

For each parameter set, flow time-series were simulated based on the full time-series of the observed climate data. For each of these flow time-series, a corresponding set of covariances (between observed climate and simulated flow) and HIs were computed. The observed covariance and HIs are also determined.

*Stage 2, evaluation:* Under the traditional approach, the hydrological model is evaluated (commonly termed validation)
following calibration using an optimisation algorithm; this presupposes that the selected hydrological model is able to capture the underlying processes (Oreskes and Belitz, 2001). The covariance approach sees the evaluation of the model structure prior to identification of the plausible parameter space. The model is invalidated, i.e. rejected, when the observed moments lie outwith the simulated moments (sampled parameter space). This may be facilitated through plots of the observed and simulated relationship between the (a) covariance between precipitation and flow, $\rho(P, Q)$, and HIs; and (b) covariance between potential
evapotranspiration and flow, $\rho(PE, Q)$, and HIs. An example for the River Nar is provided in Fig. A3. The moments may also be used to assess model equifinality (the existence of multiple behavioural parameter sets; Beven (2006); Efstratiadis and Koutsoyiannis (2010)). With a focus on evaluating the hydrological model structure, stage 2 allows consideration of the full length of the hydroclimatological time-series; split-sampling may be considered in the parameterisation of the model in stage 3. *Stage 3, parameterisation*: Selection of a model parameter set was based on a specified limit of acceptability (summarised
in Fig. 2), i.e. the ability to replicate or minimise the error (percentage difference) between the observed & simulated covariance

structures and ER HIs. In Vogel and Sankarasubramanian (2003) the focus was on the replication of a single index, whilst, in this study, the objective was the replication of multiple indices. To this end, a limit of acceptability was specified per index, with each ER HI assigned maximum error threshold based on their normalised or relative importance. The ER HI importance (Table B1) was normalised (rescaled to a range from zero to one) per catchment and the covariances assigned a relative

importance of one, equal to the most important index. The catchment specific limits of acceptability were specified as the relationship between the relative importance and a user-specified allowable error range. If no parameter sets are selected, the model structure is invalidated and rejected.

Given the large number of ER HIs identified for some catchments, an exponential model of the form $y = e^{mx+c}$ was specified for each catchment, thereby ensuring a focus on the most important indicators (see Fig. 2). In order to account for equifinality,

the maximum error was set such that the feasible parameter space was limited to approximately n = 20 distinct parameter sets (a discretionary choice made in the absence of any established rule). In Fig. 2, a simplified example is presented where the limits of acceptability are adjusted with a view to identifying a plausible parameter space where n = 3.

Note that: dependent on modelling objective, spatio-temporal transposability may be tested in stage 3 following a split-sample approach (Klemeš, 1986). As in Vogel and Sankarasubramanian (2003), the focus here is on methodological development,

thus spatio-temporal transposability is not considered.

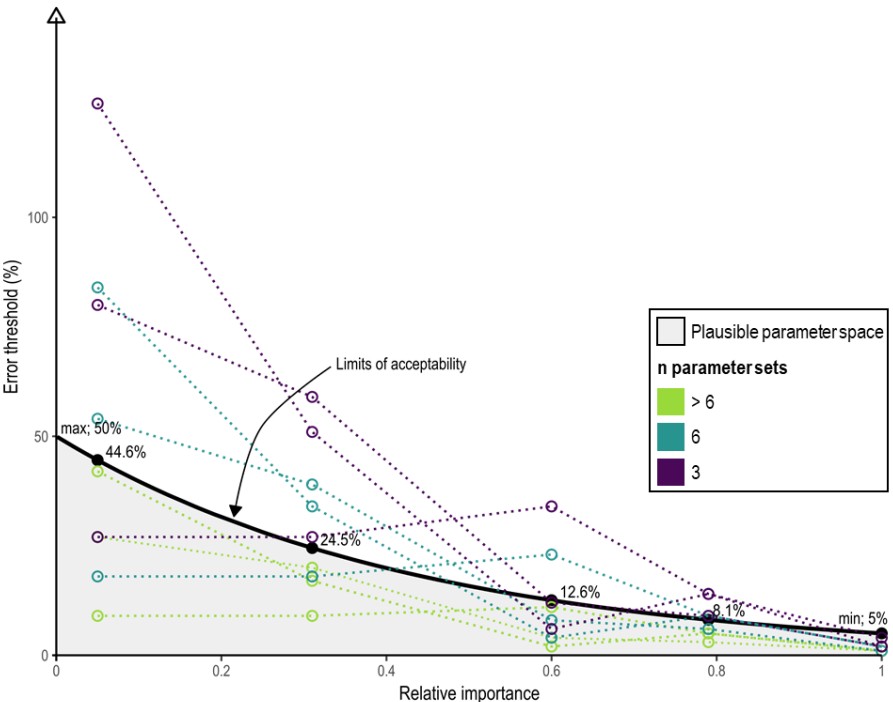

**Figure 2. Conceptualisation of the limits of acceptability, depicted here as the log-linear relationship between relative importance and the allowable (absolute) error thresholds per indicator and covariance. The limits of acceptability are reduced until *n = 3* parameter sets lie within the plausible parameter space. In this example, the error threshold ranges from 5%, where the relative**
**importance is one, to a maximum of 50%. The maximum allowable error per example indicator is marked.**

## 2.5 Model performance and consistency

In this study, the ability of the parameterised models in replicating the ecologically relevant hydrological indicators was evaluated through the evaluation metrics detailed in Table 3 (determined with reference to prior studies with similar modelling objectives: Shrestha et al. (2014); Vis et al. (2015); Pool et al. (2017)). Metrics were determined across the full time-series for each catchment~parameter set (e.g. for the River Nar, 54 years of seasonal ER HIs were determined for each of the 23 parameter sets). Three statistical tests were applied, where the goal is the rejection of the null hypothesis ($\alpha = 0.001$). Welch's t-test considers the correlation between the means of the observed and simulated indicators, whilst the KS and CvM (Cramér, 1928; Anderson, 1962) tests look to the distribution of the interquartile range and tails respectively; agreement indicates a relationship between the observed and simulated ER HIs. The hydrologic alteration factor (HAF) is adapted from the IHA approach (Mathews and Richter, 2007). It is a measure of the simulated and observed frequencies of values within three target percentile ranges: 0-25th, 25-75th, and 75-100th. As a measure of distribution, HAF is essentially a simplification of the distribution function. The acceptable range of HAF values is defined as ±0.33. Finally, two measures of error are determined: model efficiency, or the Nash-Sutcliffe Efficiency criterion (NSE), and the mean arctangent absolute percentage error (MAAPE), designed to address the limitations inherent to mean absolute relative error (Kim and Kim, 2016).

**Table 3. Descriptions, definitions and optimal values for the applied evaluation metrics. For the statistical tests, the optimal value of p < 0.001 represents the significance threshold ($\alpha = 0.001$).**

| | Metric | Description | Definition (or R-function) | Optimal value |
|---|---|---|---|---|
| *Statistical tests* | Welch's t-test | Variation on correlation where the two samples have unequal variances. Hypothesis is that two populations have equal means. | stats::t.test(…) | p < 0.001 |
| | Kolmogorov-Smirnov test (*KS*) | Tests whether samples come from the same population, i.e. follow the same distribution. | stats::ks.test(…) | p < 0.001 |
| | Cramér-von Mises (*CvM*) | Addresses limitations of KS test: (1) less focused on the central distribution; (2) more equal weighting on the tails of the distribution. | cramer::cramer.test(…) (Franz, 2014) | p < 0.001 |
| *Distribution* | Hydrologic alteration factor (*HAF*) | A factor developed as part of the Indicators of Hydrologic Alteration (Mathews and Richter, 2007). Tests the replicability of sections of the probability distribution (lower-tail, IQR and upper-tail) for a given index. | $\dfrac{F_{sim} - F_{obs}}{F_{obs}}$ Where $F$ is frequency, the no. values lying within the probability distribution. | 0 |
| *Measures of error* | Mean arctangent absolute percentage error (MAAPE) | A modification of MARE. Considers the relative error as an angle rather than a slope, reducing the bias of large errors. | $\dfrac{1}{n}\sum \arctan\left(\dfrac{I_{obs} - I_{sim}}{I_{obs}}\right)$ Where $I$ is the index value and $n$ the no. observations. | 0 |
| | Model efficiency (*NSE*) | Nash Sutcliffe efficiency. A measure of the goodness of fit of the HI to the 1:1 line (observational mean) normalised by the variance. | $1 - \dfrac{\sum(I_{obs} - I_{sim})^2}{\sum(I_{obs} - \overline{I_{obs}})^2}$ Where $I$ is the index value. | 1 |

## 3 Results

### 3.1 Model parameters

For all catchments, the low-flow optimised six-parameter GR6J model was invalidated; GR5J was invalidated for all catchments with the exception of the Tarland Burn and River Trent. A summary of the number of parameter sets (per model, per catchment) and interquartile ranges is presented Table 4 normalised (by the parameter limits specified in Table A1). For further details see Fig. B1. Being related in function, the parameters of the production ($x1$) and routing ($x3$) store capacities exhibit the greatest range. The groundwater exchange coefficient ($x4$) and inter-catchment exchange threshold ($x5$; where applicable) appear more consistent, whilst the time elapsed for the routing of flow appears inversely related to BFI.

**Table 4. Normalised interquartile (IQR) range across the parameter sets for each catchment. The average and mean values across all catchments and models are also indicated. The model GR6J was invalidated, therefore parameter x6 is omitted.**

| | Tarland Burn | | River Ribble | River Trent | | River Nar | River Thrushel | Summary | |
|---|---|---|---|---|---|---|---|---|---|
| *No. free parameters* | 4 | 5 | 4 | 4 | 5 | 4 | 4 | | |
| *No. parameter sets* | 15 | 4 | 24 | 12 | 4 | 23 | 18 | *Average* | *Median* |
| *x1* | **0.29** | **0.76** | **0.48** | **0.04** | **0.08** | **0.31** | **0.45** | 0.35 | 0.31 |
| *x2* | 0.13 | 0.05 | 0.26 | 0.11 | 0.08 | 0.04 | 0.07 | 0.10 | 0.08 |
| *x3* | **0.16** | **0.25** | **0.18** | **0.07** | **0.51** | **0.30** | **0.17** | 0.24 | 0.18 |
| *x4* | 0.09 | 0.09 | 0.03 | 0.11 | 0.04 | 0.01 | 0.02 | 0.06 | 0.04 |
| *x5* | - | 0.05 | - | - | 0.08 | - | - | 0.06 | 0.06 |

### 3.2 Model performance and consistency

The ability of the covariance approach in the replication of the ER HIs is considered in terms of performance and consistency. The models are evaluated with reference to the metrics summarised in Table 3 previously. Results are considered by metric, with a focus on the ER HIs with the best and worst performance and consistency.

### 3.2.1 Statistical tests

A series of tests were applied with a view to determining if, statistically speaking, the observed and simulated ER HIs come from the same population. The tests focus on the mean (t-test), the central distribution (KS) and tails of the distribution (CVM test). Table B1 in the appendix details, per ER HI and catchment, the percentage of the parameter sets which did not show a significant level of agreement.

The statistical tests saw perfect agreement across all six timing indicators. With respect to the magnitude indices, the ER HI *BFIr* and the three skewness indicators do not satisfy any of the tests; performance appears irrespective of importance indicated by the hydroecological model or catchment. Magnitude median-quantile ratios agreement was mixed, with high and low flows achieving poor and good agreement respectively. Broadly, frequency indicators indicate a lack of agreement, with only the *PlsFld* index in the River Thrushel exhibiting performance and consistency. The role of statistical importance in the replication

of these more complex indicators is also suggested, with *PlsQ75* replicated well in the Tarland (importance 0.69) and poorly in the Trent (importance 0.03). More broadly, log-transformed indicators saw better agreement; for example, the more important *MaxMonthlyVar* generally performed poorly, whilst *MaxMonthlyLogVar* saw agreement across all tests and parameter sets.

### 3.2.2 Distribution – Hydrologic alteration factor (HAF)

The hydrologic alteration factor (HAF) is a test of the replicability of the shape of the probability distribution. Fig. 3 summarises the HAF value across the central distribution and tails for each ER HI. There is agreement across the percentile ranges for the majority of the ER HIs considered. Notably, the 19 (of 22; statistics, log-ratios and quantile-median ratios) magnitude indicators not pictured achieved optimal HAF of zero. The three-monthly indicators (depicted) again highlight relative success in replicating a log-transformed index.

The performance of the six indicators capturing flow pulse events is varied: the central distribution of flood pulses is well-replicated whilst the upper tail exhibits a consistent large negative bias. The HAF values also serve to highlight some inconsistencies in the performance of the timing indicators. A variable negative bias is in evidence for the index *Mn7MaxJD*, however, in this case, it is worth noting that it is inherently more difficult for a hydrological model to detect and replicate (multiple) short-term events (Pool et al., 2017). Perhaps surprisingly, *Mn90MnJD* is subject to a large positive bias in the lower tail, i.e. the range of the distribution is underestimated. In contrast to *Mn7MaxJD*, this discrepancy may be due to the long(er)-term duration; with seasons of approximately 180 days in length, there are a limited number of values the indicator can take.

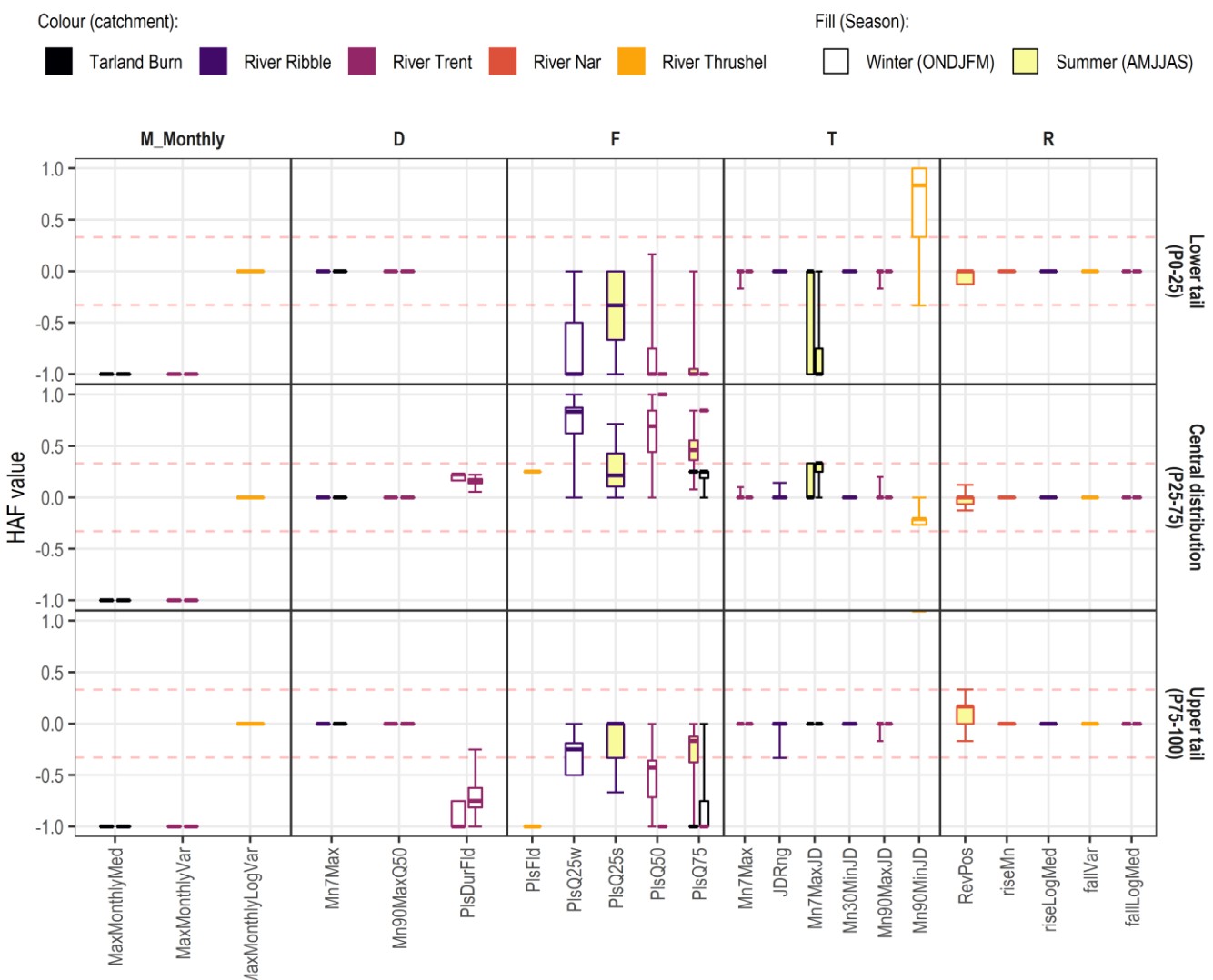

**Figure 3. Hydrologic alteration factor (HAF) values for the three percentile ranges for each ER HI; ER HIs are grouped by facet of the flow regime: magnitude (M), duration (D), frequency (F), timing (T) and rate of change (R). The acceptable range of HAF values is defined as ±0.33 (red dashed line); HAF > 0 represents an increase in frequenecy relative to the observed whilst HAF < 0 represents a decrease. All magnitude statistic and ratio ER HIs achieved optimal values (HAF = 0) and are not depicted. The 4- and 5-parameter results are adjacent, left and right respectively, for the Tarland Burn and River Trent.**

### 3.2.3 Error – MAAPE and NSE

Two measures of error were applied, MAAPE, a modification of the mean absolute relative error (MARE) which reduces the bias of large errors, as well as the more commonplace Nash Sutcliffe efficiency (NSE). The MAAPE for each ER HI is depicted in Fig. 4; to ensure consistency with HAF, acceptable boundaries are specified as ±0.33 (depicted, horizontal red lines). Overall, the same general patterns may be observed; for example, skew indicators are not well replicated, log-transformation improves the monthly index performance, and timing, with the exception of *Mn90MinJD*, achieves consistently good

performance. However, it is clear that the consideration of multiple parameter sets per catchment model leads to variation in the simulated ER HI which may not have been detected by the previous metrics. MAAPE also serves to highlight the difference in performance across the median-quantile ratios, extreme high-flow indices (*Qmax* to *Q05*) are over-estimated whilst the replication of low-flow indices is subject to considerably less (negative) bias.

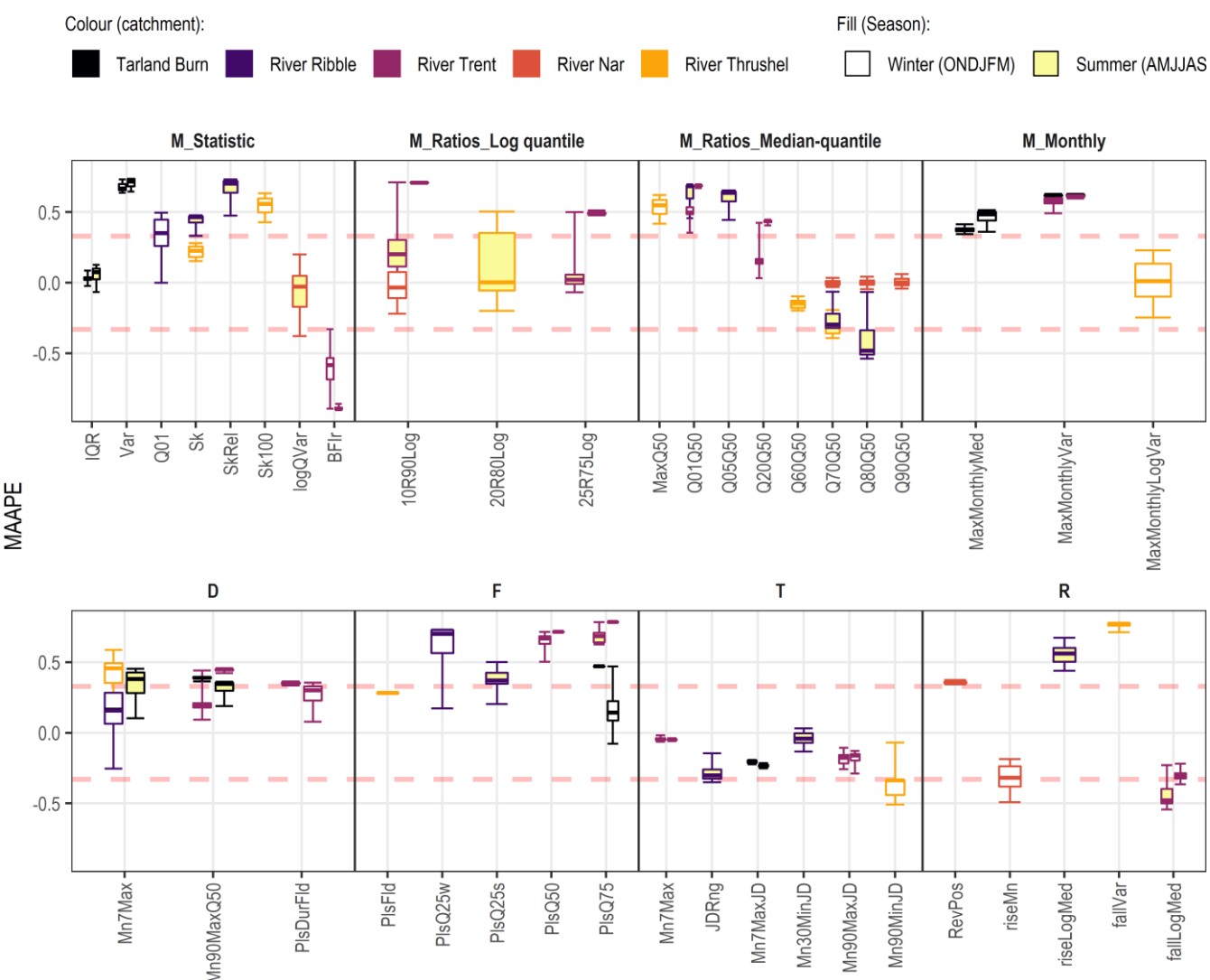

**Figure 4. Mean arctangent absolute percentage error (MAAPE) values for each ER HI; ER HIs are grouped by facet of the flow regime: magnitude (M), duration (D), frequency (F), timing (T) and rate of change (R). As per HAF, the acceptable range is defined as ±0.33 (red dashed line). The 4- and 5-parameter results are adjacent, left and right respectively, for the Tarland Burn and River Trent.**

The NSE is a measure of model efficiency where values less than zero suggests that the observational mean may be a better estimate. In Fig. 5, only ER HI with NSE > 0 are depicted with the number of parameter sets described as *n*; for all ER HI see Fig. B2.

Seventeen ER HI achieved NSE values greater than zero; further, the low values of n which are in evidence (Fig. 5) indicate a lack of consistency across parameter sets. Those ER HI which have already been shown to perform well are indicated, examples include the low flow median-quantile ratios, the log-transformed monthly index and the timing indicators more generally.

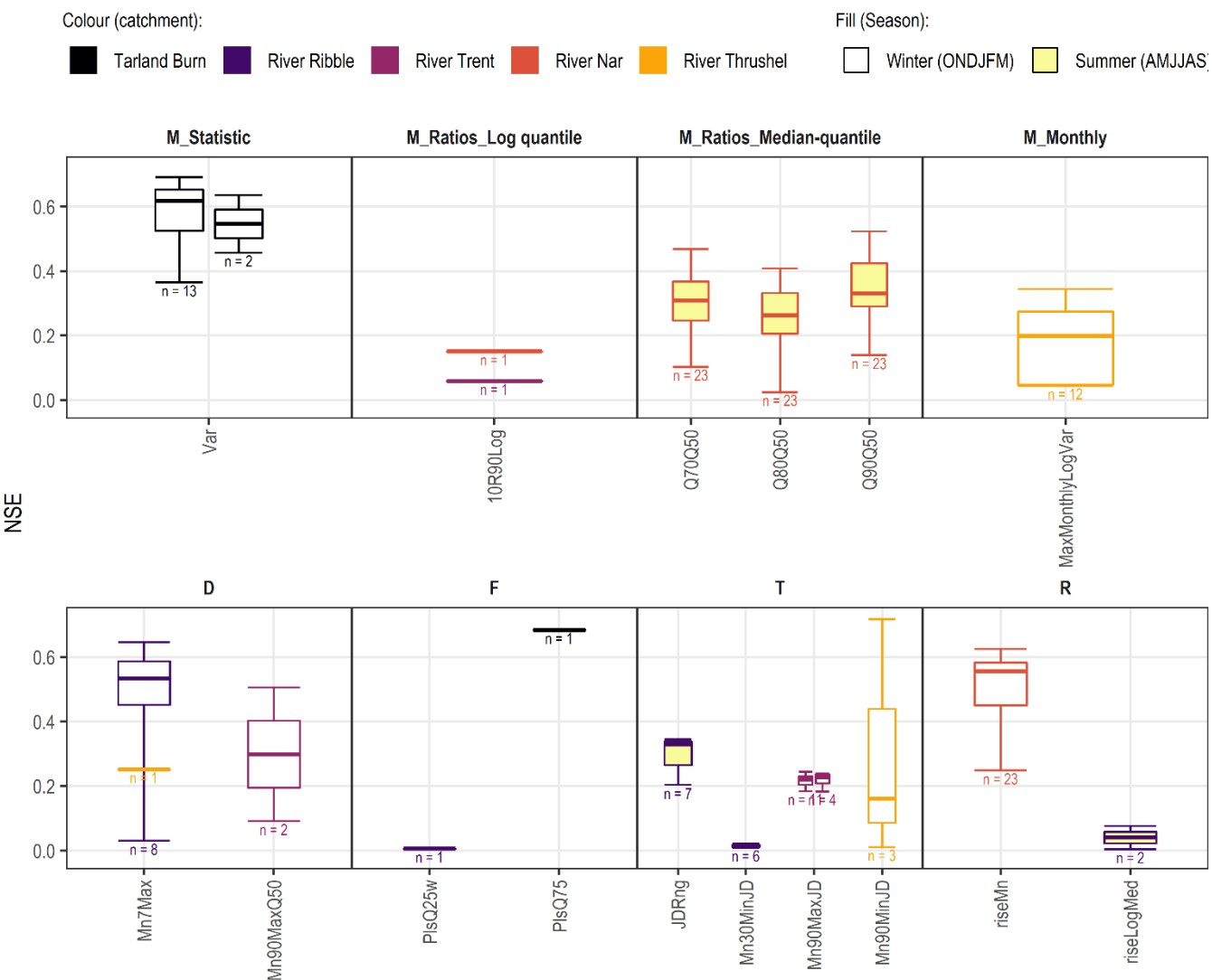

**Figure 5. Nash Sutcliffe Efficiency (NSE) for each ER HI where NSE > 0 (model skill greater than observational mean); see Fig. B2 for all NSE. The ER HIs are grouped by facet of the flow regime: magnitude (M), duration (D), frequency (F), timing (T) and rate of change (R). The 4- and 5-parameter results are adjacent, left and right respectively, for the Tarland Burn and River Trent.4 Discussion**

There is a clear need to understand the impact of hydrologic change on the river ecosystem. To this end, hydrological models are used to simulate flow time-series from which ecologically relevant hydrological indicators are derived. Previous studies

(e.g. Vis et al. (2015), Shrestha et al. (2014) and Pool et al. (2017)) have highlighted the inability of hydrological models to simulate a range, or suite, of ER HIs concurrently. In this study, a modification of Vogel and Sankarasubramanian (2003) covariance approach was applied to five hydrologically distinct catchments; the focus was on the replication of a suite of ER HIs identified through catchment-specific hydroecological models. The ability of this modified covariance approach, in terms of performance and consistency, was assessed through a series of evaluation metrics.

A range of catchments was, with the main differences lying in the catchment BFI, length of the available time-series and the ER HIs. In this study, BFI ranged from 0.25 to 0.91, essentially flashy to groundwater-fed. With the exception of model parameterisation, there was no discernible difference in the replication of ER HIs. Similarly, the length of the available time-series appears to have made no observable difference to the replicability of the ER HI distributions specifically. In terms of error, MAAPE and NSE, lower overall performance for the shorter time-series is expected as a result of sample size sensitivity. Finally, despite consideration of a range of ER HIs with different associated importance, there appears a consistent message in terms of the performance and consistency of similar indices and the facets of the flow regime more broadly.

## 4.1 Performance and consistency

The consideration of a range of catchments provides a clear picture of the capacities of the hydrological models as well as the relative success of the covariance approach. Overall, replication of the ER HIs was good. Timing and log-transformed indicators (*logQVar*, *MaxMonthlyLogVar* and the log quantile ratios) were among the most consistent and well-replicated across the range of catchments. The results are broadly consistent with a number of recent studies (Melsen et al., 2018; Mackay et al., 2019; Worthington et al., 2019) where timing and duration indicators are among the indicators with the highest prediction accuracy. Difficulties were observed in replicating frequency and rate of change indices. Replication of indicators incorporating the seasonal median flow (Q50) was also poor, with large positive biases frequently observed. This may be observed directly through comparison of the replication of *Q01* and *Q01Q50* in the River Trent where the degree of error can be seen to markedly increase. Recent studies by Mackay et al. (2019) and Worthington et al. (2019) also observed higher error rates for monthly indicators.

### 4.1.1 Suitability of ER HIs in hydrological modelling

This, and previous studies, have observed difficulties in the replication of frequency ER HIs (flow pulses). This begs the question: Is this a product of the covariance approach? An inherent limitation of hydrological models more generally? Or is this related to the nature of the indicator itself? A review of the simulated flow suggests the latter. There is a tendency for the simulations to identify shorter more frequent pulses, whilst the observed pulses are longer and less frequent. For instance, the median error (MAAPE) for *PlsQ50* (the number of pulses above a baseline Q50 threshold) on the River Trent was 0.75; this falls to 0.368 if the focus is on the total duration of the pulses. The pooling of events with an inter-event time below some threshold, as per the inter-event time and volume criterion (Gustard and Demuth, 2009) for example, may serve to improve

the replication of the pulse indicators. It should be noted that, in this study, this limitation does not extend to flood pulses (*FldPls*) due to the much larger inter-event time, thus allowing for better replication of flood pulses overall.

In multiple cases, this study observed difficulties in replicating those ER HIs which are considered relative to the median seasonal flow. Comparison of the indicators *Q01* and *Q01Q50* in the same catchment indicates that the lack of direct consideration of median flows in the parameterisation of the model may be a limiting factor. Indeed, it may be that the decomposition of such indicators into their component parts, e.g. Q01 and Q50, may lead to better replicability overall. Similarly, the results indicate that log-transformation of flows may lead to improvements in the replicability of certain ER HIs. Further work is required to confirm this premise.

### 4.1.2 Suitability of evaluation metrics

There is a lack of consistency in the evaluation metrics considered in the evaluation of hydrological model performance. Further, these studies make use of metrics which exhibit known bias, for example, mean absolute relative error (MARE; Kim and Kim (2016) and NSE (Gupta et al., 2009; Pushpalatha et al., 2012; Vis et al., 2015). For the measure of error, this study replaced the former with MAAPE (see Table 3). The reasons for the consideration of NSE in this study were twofold: (1) application of NSE is the norm; and (2) to illustrate the limitations of this measure. The limitations of NSE are frequently cited as low scores where there is high variability (Gupta et al., 2009) as well as a bias towards high flows (Pushpalatha et al., 2012). Additionally, the NSE is scaled by the standard deviation, rendering it incomparable across catchments (Gupta et al., 2009). In this study, only seventeen of the ER HIs achieved NSE > 1, i.e. the simulations are better than an estimation based on the observed mean. Similar observations were made in Vis et al. (2015). It can be concluded that, given this lack of robustness, NSE is not a suitable evaluation metric in studies such as this one.

### 4.2 Advantages and limitations of the modified covariance approach

In this section we consider the general advantages of the modified covariance approach, relative to the traditional approach; this is followed by consideration of the hydroecological modelling requirements. It is clear that no approach has been able to achieve adequate performance and consistency in the replication of more complex ER HIs, specifically those related to rate of change. Shrestha et al. (2014) observed difficulties in replicating low flows, the duration of flow pulses, and monthly flows specifically. In this study, no such observations have been made with regards to low flows and duration, indeed, these may be considered to be relatively well-replicated across all catchments. Poor replication of monthly ER HIs does however persist; log-transformed variations of these indicators may represent a viable alternative. Whilst Pool et al., 2017 saw improvements (relative to Shrestha et al. (2014) and Vis et al. (2015)), the need to calibrate the model to each ER HI in question would strongly call into question the reliability of the hydrological model (due to the inability of the hydrological model to simulate catchment hydrological processes simultaneously). The consistency with which (the majority of the) ER HIs are replicated here illustrates that this is not a necessary limitation of hydrological models. A lack of consistency in ER HIs demonstrating

elevated levels of variability, such as high flows, is to be expected due to the dynamic nature of inter-annual weather patterns (Pool et al., 2017).

### 4.2.1 General advantages

Here follows a brief discussion of the general advantages of the modified covariance approach. First, uncertainty is reduced via a number of avenues:

- *Disinformative data:* Models calibrated following a traditional approach are particularly sensitive to measurement error (Westerberg et al., 2011). Lack of agreement in the observed-simulated time-series, even for a single event, may bias the objective function, leading to rejection of an otherwise well-performing parameter set (Beven, 2010; Westerberg et al., 2011). Methods which do not focus on the replication of time-series directly, such as the modified covariance approach, are known to limit the influence of input uncertainty (Westerberg et al., 2011; Euser et al., 2013);

- *Validation of model structure:* Consideration of the observed and simulated moments allows the user to evaluate the ability of the hydrological model structure in capturing the hydrological processes in the catchment, thus ensuring the selection of the optimal model (structure);

- *Equifinality:* Equifinality, reaching the same outcome by different means, is a major challenge of hydrological modelling. In the modified covariance approach the entire parameter space is considered at the outset. A plausible parameter space is determined by focussing on the region which is best able to replicate the characteristics of the HIs, thereby reducing the epistemic uncertainty associated with accounting for equifinality (Wu et al., 2017).

Finally, whilst the large number of simulations required under the modified covariance approach may seem prohibitive, this demand may be offset. Unlike the traditional approach, where selection algorithms may introduce issues of speed and accuracy (Seibert, 2000), finite time is needed to apply the covariance approach. All simulations of the hydrological model are performed at the outset; once the full suite of parameter sets have been simulated the hydrological model need not be run again. Under a more traditional approach, such as in (Pool et al., 2017) where the ER HIs serve as the objective, the HIs must be specified at the outset. This is not the case in the modified covariance approach, where the *n* Monte Carlo simulations can be performed in advance of HI selection. Thus, multiple suites of ER HIs may be considered (e.g. all rate of change or magnitude indicators) with limited additional time outlay.

### 4.2.2 Hydroecological model requirements

The explicit consideration of the outcomes of hydroecological modelling is perhaps both the most significant advantage and disadvantage of the modified covariance approach. Whilst hydrological modelling informed by the outcomes of hydroecological studies is not new, for instance, Pool et al. (2017) was informed by Knight et al. (2014), the novelty of this approach lies in the explicit consideration of the statistical importance of the ER HIs, identified through hydroecological modelling. The consideration of the relative importance of each ER HI allows a large suite of ER HIs (seven to thirteen) to be considered with no apparent penalties. Further, contrary to expectations, a large number of important ER HIs ($> 0.5$) has no impact on replicability. In the

case of the River Ribble, where a total of thirteen ER HIs were considered, seven had an importance greater than 0.5. Similarly, through this approach, a high weighting is not needlessly attributed to ER HIs with low importance.

The need for a hydroecological model represents the major limiting factor due to the requirement for long-term hydroecological time-series. Historically, hydrological and ecological data were collected for different objectives (Poff and Allan, 1995; Knight et al., 2008; Monk et al., 2008), leading to a mismatch in temporal and spatial coverage. High levels of disparity in sampling and gauging sites inevitably introduce noise into the model. However, the availability of national ecological datasets, such as BIOSYS in the UK, may serve to offset the issue of data availability. Such datasets may be used to develop regional hydroecological models based on flow regime type and the assumption of homogeneity in environmental conditions. The modified covariance approach may also be applied without a numerical measure of the relative importance of each indicator, this would however introduce an element of subjectivity into the parameterisation of the model.

## 4.3 Wider applicability and further work

The modified covariance approach is able to provide statistically robust simulations and projections of ER HIs for applications such as environmental flow assessment or in assessing the hydroecological impact of climate change such as in Visser et al. (2019a) and Visser et al. (2019b). However, the applicability of the approach may not be limited to hydroecological studies and the simulation of ER HIs (e.g. replication of hydrological signatures). In this context, example applications could include the replication of water resource management indicators (monthly, seasonal and annual flows). Such applications would require consideration of a statistical model for the determination of the statistical importance of indicators. The approach may also be used in the development of regional hydrological models, thereby facilitating the simulation of ER HIs in ungauged catchments. Finally, the clarity with which model structures are accepted or rejected makes the approach apt for use in combination with model selection frameworks such as the Framework for Assessing the Realism of Model Structures (FARM; Euser et al. (2013)).

## 5 Concluding remarks

This study considered the performance and consistency of a modified covariance approach in the replication of ecologically relevant hydrological indicators. Application across five hydrologically diverse catchments showed a consistent level of performance across the majority of ER HIs; the timing facets of the flow regime were best replicated, whilst rate of change indicators saw the poorest performance and consistency. Relative to similar studies, there was an overall improvement in consistency, thus, this study represents an important advancement towards the robust application of hydrological models for ecological flow studies. The explicit consideration of hydroecological modelling outcomes allows the hydrological model to be tuned to parameters based on statistical importance. A further major advantage of the modified covariance approach lies in the identification of the plausible parameter space which best captures (replicates) the characteristics of the ER HIs, thereby providing a greater understanding of the suitability, limitations and uncertainties of the hydrological model structure.

*Data availability:* The hydroclimatological data used for all catchments (except the Tarland Burn) is freely available from the NRFA (2018), Met Office (2018a) and Met Office (2018b). Data for the Tarland Burn was provided to Heriot-Watt on request for this study by the James Hutton Institute (JHI, 2018).

*Author contributions:* AV developed the methodology, code and performed the data analysis. AV prepared the manuscript whilst LB provided review and edits. Both LB and SP provided supervision.

*Competing interests.* The authors declare that they have no conflict of interest.

*Acknowledgements:* The authors gratefully acknowledge funding from the Engineering and Physical Science Research Council through award 1786424.

**Appendix A – Method**

**A.1 Case studies**

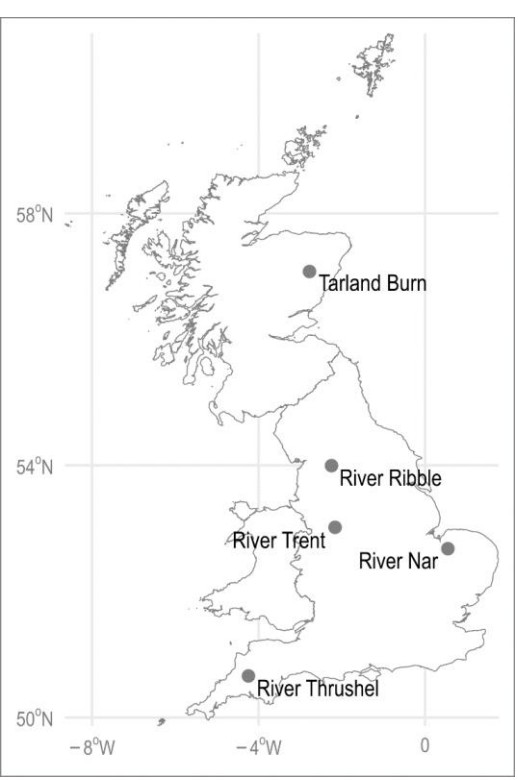

**Figure A1. Distribution of the case study catchments across the UK.**

## A.2 Hydroecological modelling

Based on Olden and Poff (2003) and Monk et al. (2006), daily mean flow data was used to derive 63 hydrological indices per hydrological season: winter (ONDJFM) and summer (AMJJAS); for data source, see Table 1. Principal Component Analysis (PCA) was applied to identify those indices which describe the major aspects of the flow regime whilst minimising redundancy.

Macroinvertebrates serve as the proxy for ecological response. Response is determined using the Lotic-Invertebrate Index for Flow Evaluation, accounting for macroinvertebrate flow velocity preferences (Extence et al., 1999). For four out of five case studies LIFE scores were determined to family level; data for the River Nar, obtained directly from the Environment Agency, was available to species level. The modelling focused on spring ecological activity (the period of peak activity and largest consistent availability of data).

After Visser et al. (2019b), an Information Theory approach to modelling was taken in order to provide a quantitative measure of support for parameters and candidate models. Inference is made from multiple models through model averaging. In summary: (1) the candidate models are evaluated with respect to the second-order bias corrected Akaike Information Criterion (AICc) (after Burnham and Anderson (2002); see also Visser et al. (2019b)); (2) a best approximating model is inferred from a weighted combination of all the candidate models; (3) the parameters are ranked, such that the highest value represents the

most important in the model; (4) filters are applied to remove parameters where the estimate and confidence intervals are zero (i.e. certainty that the index is not to be included) and to reduce the model to the parameters which describe 95% of the cumulative information. For further details, see Visser et al. (2018) and Visser et al. (2019b).

## A.3 Hydrological modelling

**Table A1. Parameter limits for the hydrological models.**

|    | Description | Limits |
|----|------------------------------------------------------------|------------|
| x1 | Capacity of production store (mm) | (100,1200} |
| x2 | Groundwater transfer (mm/day; positive indicates flow *from* aquifer) | (-5,25} |
| x3 | Capacity of routing store (mm) | (20,1000} |
| x4 | Time lag between rainfall event and flow (days) | (0.5,30} |
| x5 | Inter-catchment exchange threshold (-) | (-5,25} |
| x6 | Capacity of parallel routing store (mm) | (20,1000} |

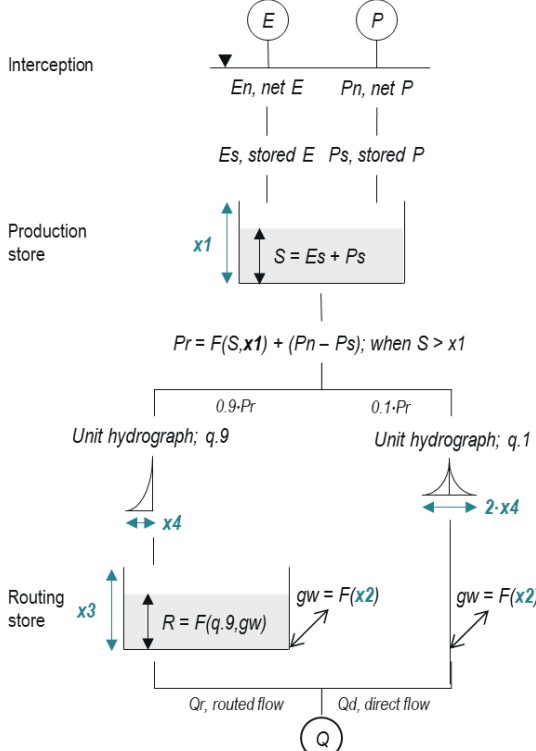

**Figure A2. Structure of the GR4J hydrological model; based on Perrin et al. (2003). The 5-parameter GR5J sees the addition of x5, inter-catchment exchange parameter, at the same locations as x2, whilst GR6J sees the addition of a store parallel, capacity x6, to the routing store.**

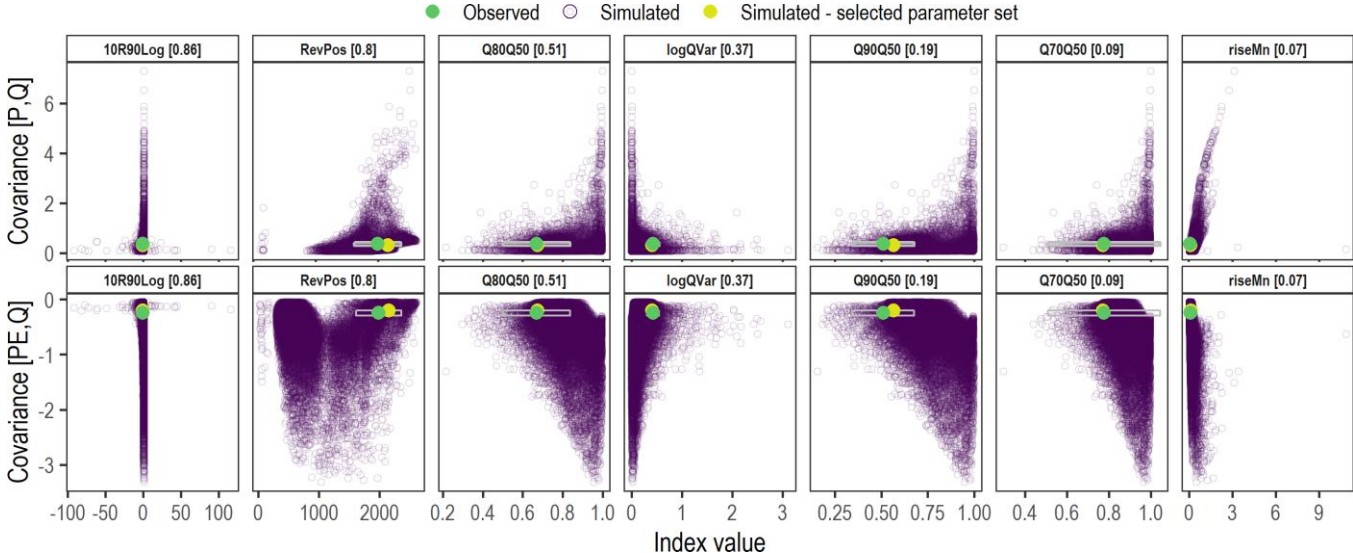

**Figure A3. Observed and simulated moments for the 100,000 Monte Carlo simulations using the GR4J model for the River Nar case study. The grey boxes depict the boundaries of the limits of acceptability per index. One of the selected parameter sets, $i = 73,952$, is highlighted (yellow).**

# Appendix B – Supplementary results

## B.1 Ecologically relevant hydrological indices and test statistics

Table B1. Ecologically relevant hydrological index descriptions; grouping is by facet of the flow regime. Seasons are indicated through no shading (winter) and shading (summer). Subsequent columns are catchment specific, denoting ER HI importance, and the results of the statistical tests detailed in Table 3. In the table, a flood threshold is the flow equivalent for a flood recurrence interval of 1.67 years (on the baseline).

*Four and five parameter models were applied to both the Tarland Burn and River Trent. Single digit entries should be interpreted as being the same across both models; where entries are separated, e.g. for *10R90Log*, the former represents GR4J and the latter GR5J.

| Index | Description | Units | Tarland Burn* Importance | Normal | Mean | IQR | Tails | River Ribble Importance | Normal | Mean | IQR | Tails | River Trent* Importance | Normal | Mean | IQR | Tails | River Nar Importance | Normal | Mean | IQR | Tails | River Thrushel Importance | Normal | Mean | IQR | Tails |
|---|---|---|---|---|---|---|---|---|---|---|---|---|---|---|---|---|---|---|---|---|---|---|---|---|---|---|---|
| *Magnitude - Statistic* | | | | | | | | | | | | | | | | | | | | | | | | | | | |
| IQR | Interquartile range of flow. | $m^3s^{-1}$ | 0.43 | 0 | 0 | 0 | 0 | | | | | | | | | | | | | | | | | | | | |
| Var | Variance in flow. | - | 0.46 | 0 | 0 | 0 | 0 | | | | | | | | | | | | | | | | | | | | |
| Q01 | Q1 flow (extreme high flow). | $m^3s^{-1}$ | | | | | | 0.32 | 0 | 83.3 | 70.8 | 87.5 | | | | | | | | | | | | | | | |
| Sk | Skewness, mean relative to median. | - | | | | | | 0.11 | 0 | 100 | 100 | 100 | | | | | | | | | | | 0.88 | 0 | 100 | 100 | 100 |
| SkRel | Relative skewness, mean minus median, relative to median. | - | | | | | | 0.09 | 0 | 100 | 100 | 100 | | | | | | | | | | | | | | | |
| Sk100 | Range relative to the median. | - | | | | | | | | | | | | | | | | | | | | | 0.38 | 0 | 100 | 100 | 100 |
| logQVar | Variance in log-transformed flow. | - | | | | | | | | | | | | | | | | 0.37 | 0 | 0 | 0 | 0 | | | | | |
| BFIr | The seasonal BFI relative to baseline BFI. | - | | | | | | | | | | | 0.03 | 0 | 100 | 100 | 100 | | | | | | | | | | |

*Magnitude - Ratios - Log quantile*

| Metric | Description | | A1 | A2 | A3 | A4 | A5 | B1 | B2 | B3 | B4 | B5 | C1 | C2 | C3 | C4 | C5 | D1 | D2 | D3 | D4 | D5 |
|---|---|---|---|---|---|---|---|---|---|---|---|---|---|---|---|---|---|---|---|---|---|---|
| 10R90 Log | | | | | | | | 0.97 | 100 | 0 | $\frac{8.3}{100}$ | 0 | 0.86 | 100 | 0 | 0 | 0 | | | | | |
| 20R80 Log | Log-transformed ratio, $xx^{th}$ to $yy^{th}$ percentile flow. | - | | | | | | | | | | | | | | | | 0.94 | 100 | 0 | 0 | 0 |
| 25R75 Log | | | | | | | | 0.97 | 100 | 0 | $\frac{8.3}{100}$ | 0 | | | | | | | | | | |

*Magnitude - Ratios - Median-quantile*

| Metric | Description | | A1 | A2 | A3 | A4 | A5 | B1 | B2 | B3 | B4 | B5 | C1 | C2 | C3 | C4 | C5 | D1 | D2 | D3 | D4 | D5 |
|---|---|---|---|---|---|---|---|---|---|---|---|---|---|---|---|---|---|---|---|---|---|---|
| MaxQ50 | Max. flow relative to median (extreme high flow). | - | | | | | | | | | | | | | | | | 0.32 | 0 | 100 | 100 | 100 |
| Q01Q50 | | | 0.53 | 0 | 100 | 100 | 100 | 0.03 | 0 | 100 | 100 | 100 | | | | | | | | | | |
| Q05Q50 | $Qxx$ flow relative to median (high flow). | - | 0.4 | 0 | 100 | 100 | 100 | | | | | | | | | | | | | | | |
| Q20Q50 | | | | | | | | 0.03 | 0 | $\frac{16.7}{100}$ | $\frac{66.7}{100}$ | $\frac{58.3}{100}$ | | | | | | | | | | |
| Q60Q50 | | | | | | | | | | | | | | | | | | 0.97 | 0 | 72.2 | 33.3 | 72.2 |
| Q70Q50 | $Qxx$ flow relative to median (low flow). | - | 0.88 | 0 | 83.3 | 66.7 | 83.3 | | | | | | 0.09 | 0 | 0 | 0 | 0 | 0.99 | 0 | 77.8 | 55.6 | 72.2 |
| Q80Q50 | | | 0.38 | 0 | 83.3 | 75 | 87.5 | | | | | | 0.51 | 0 | 0 | 0 | 0 | | | | | |
| Q90Q50 | | | | | | | | | | | | | 0.19 | 0 | 0 | 0 | 0 | | | | | |

*Magnitude - Monthly*

| Metric | Description | Units | | | | | | | | | | | | | | | | | | | | | | |
|---|---|---|---|---|---|---|---|---|---|---|---|---|---|---|---|---|---|---|---|---|---|---|---|---|
| Max Monthly Med | Median of max. monthly flow. | $m^3s^{-1}$ | 0.7 | 0 | 0/25 | 0 | 0 | | | | | | | | | | | | | | | | | |
| Max Monthly Var | Variability in max. monthly flow. | - | 0.45 | 0 | 0 | 0 | 0 | | | | | | 0.92 | 100 | 33.3/100 | 91.7/100 | 91.7/100 | | | | | | |
| Max Monthly LogVar | Variability in max. monthly log-transformed flow. | - | | | | | | | | | | | | | | | | | | 0.45 | 100 | 0 | 0 | 0 |

*Duration*

| Metric | Description | Units | | | | | | | | | | | | | | | | | | | | | | |
|---|---|---|---|---|---|---|---|---|---|---|---|---|---|---|---|---|---|---|---|---|---|---|---|---|
| Mn7 Max | Mean of the 7-day cumulative max. flow. | $m^3s^{-1}$ | 0.53 | 0 | 0 | 0 | 0 | 0.14 | 0 | 20.8 | 33.3 | 25 | | | | | | | 0.5 | 0 | 94.4 | 77.8 | 88.9 |
| Mn90 MaxQ50 | Mean of the 90-day cumulative max. flow relative to the median. | - | 0.53 | 0 | 0 | 0 | 0 | | | | | | 0.06 | 0 | 25/100 | 16.7/100 | 33.3/100 | | | | | | |
| PlsDur Fld | Duration of pulses above a (baseline) flood threshold. | Days | | | | | | | | | | | 0.02 | 100 | 0 | 0 | 50/25 | | | | | | |
| PlsDur Q75Var | Variation in the duration of pulses below a Q75 (baseline) threshold. | - | | | | | | | | | | | | | | | | | | | | | | |

*Frequency*

| Code | Description | Units | | | | | | | | | | | | | | | | | | | | | | | | |
|---|---|---|---|---|---|---|---|---|---|---|---|---|---|---|---|---|---|---|---|---|---|---|---|---|---|---|
| PlsFld | No. of pulses above a (baseline) flood threshold. | Count | | | | | | | | | | | | | | | | | | | | 0.41 | 100 | 0 | 0 | 55.6 |
| PlsQ25w | No. of pulses above a Q*xx* (baseline) threshold. | Count | | | | | | 0.64 | 0 | 91.7 | 83.3 | 95.8 | | | | | | | | | | | | | | |
| PlsQ25s | | | | | | | | 0.58 | 0 | 66.7 | 70.8 | 83.3 | | | | | | | | | | | | | | |
| PlsQ50 | | | | | | | | | | | | | 0.04 | 0 | 100 | 100 | 100 | | | | | | | | | |
| PlsQ75 | No. of pulses below a Q*xx* (baseline) threshold. | Count | 0.69 | 0 | 0 | 0 | 0 | | | | | | 0.03 | 0 | 100 | 100 | 100 | | | | | | | | | |

*Timing*

| Code | Description | Units | | | | | | | | | | | | | | | | | | | | | | | | |
|---|---|---|---|---|---|---|---|---|---|---|---|---|---|---|---|---|---|---|---|---|---|---|---|---|---|---|
| JDRng | Range in the Julian days for the max. and min. daily mean flow. | JD | | | | | | 0.73 | 0 | 0 | 0 | 0 | | | | | | | | | | | | | | |
| Mn7 MaxJD | Julian day of the mean 7-day max. flow. | JD | 0.78 | 0 | 0 | 0 | 0 | | | | | | 0.94 | 0 | 0 | 0 | 0 | | | | | | | | | |
| Mn30 MinJD | Julian day of the mean 30-day min. flow. | JD | | | | | | 0.67 | 0 | 0 | 0 | 0 | | | | | | | | | | | | | | |
| Mn90 MaxJD | Julian day of the mean 90-day max. flow. | JD | | | | | | | | | | | 0.03 | 100 | 0 | 0 | 0 | | | | | | | | | |
| Mn90 MinJD | Julian day of the mean 90-day min. flow. | JD | | | | | | | | | | | | | | | | | | | | | 0.88 | 100 | 0 | 0 | 0 |

*Rate of change*

| | | | | | | | | |
|---|---|---|---|---|---|---|---|---|
| RevPos | No. days when flow increases (positive reversals). | Days | | | 0.8 | 0 | 100 | 100 | 100 |
| riseMn | Mean rise rate (flow increasing). | m$^3$s$^{-1}$ | | | 0.07 | 0 | 0 | 0 | 0 |
| riseLog Med | Median log-transformed rise rate (flow increasing). | m$^3$s$^{-1}$ | 0.55 0 0 0 4.17 | | | |
| fallVar | Variation in fall rate (flow decreasing). | - | | | | 0.16 0 100 100 100 |
| fallLog Med | Median log-transformed fall rate (flow decreasing). | m$^3$s$^{-1}$ | | 0.93 100 — 100 (91.7 75) | | |

## B.2 Model parameters

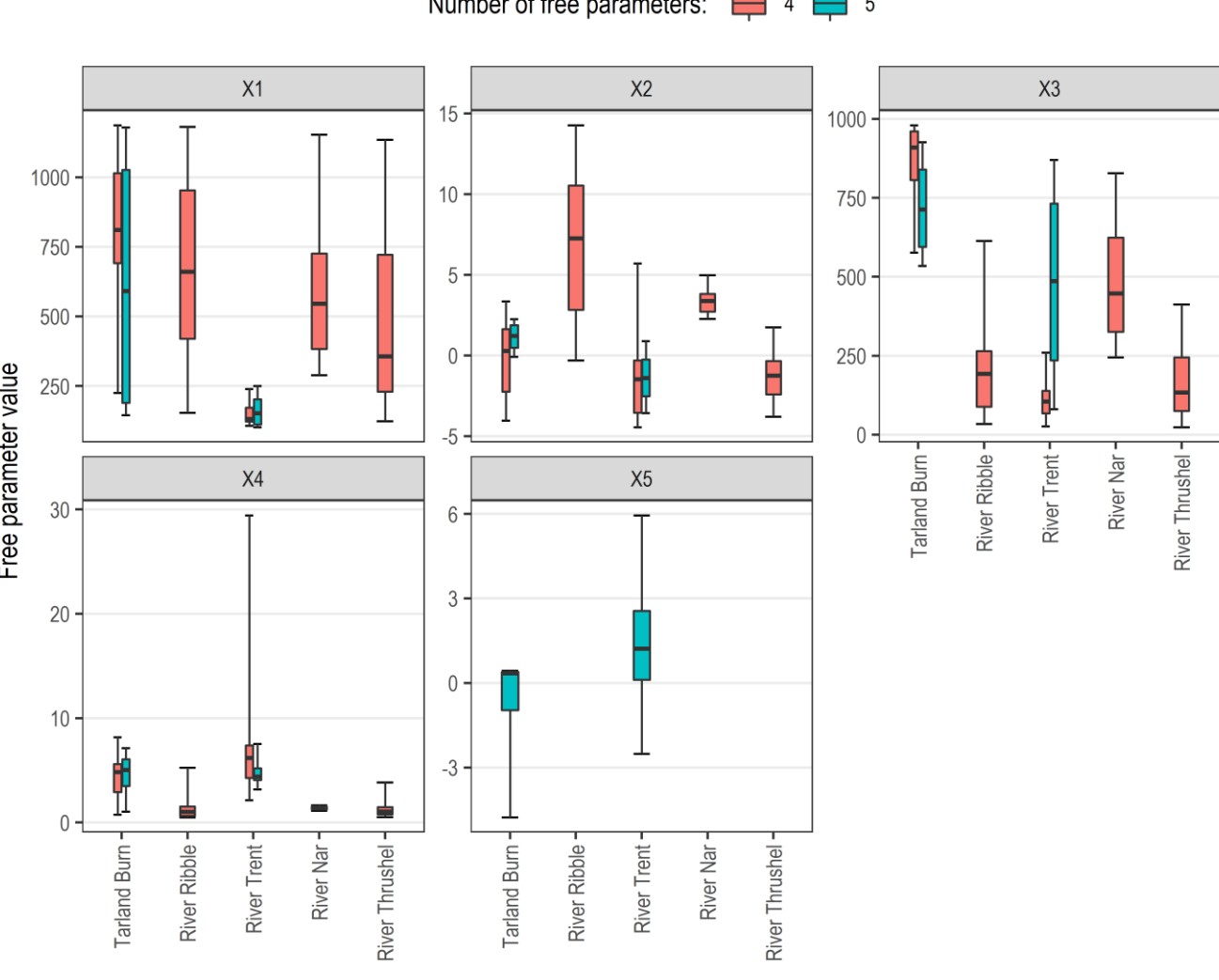

**Figure B1. Boxplots of the parameter values across the 100 selected models. The whiskers represent the maximum and minimum values observed.**

## B.3 Nash Sutcliffe Efficiency

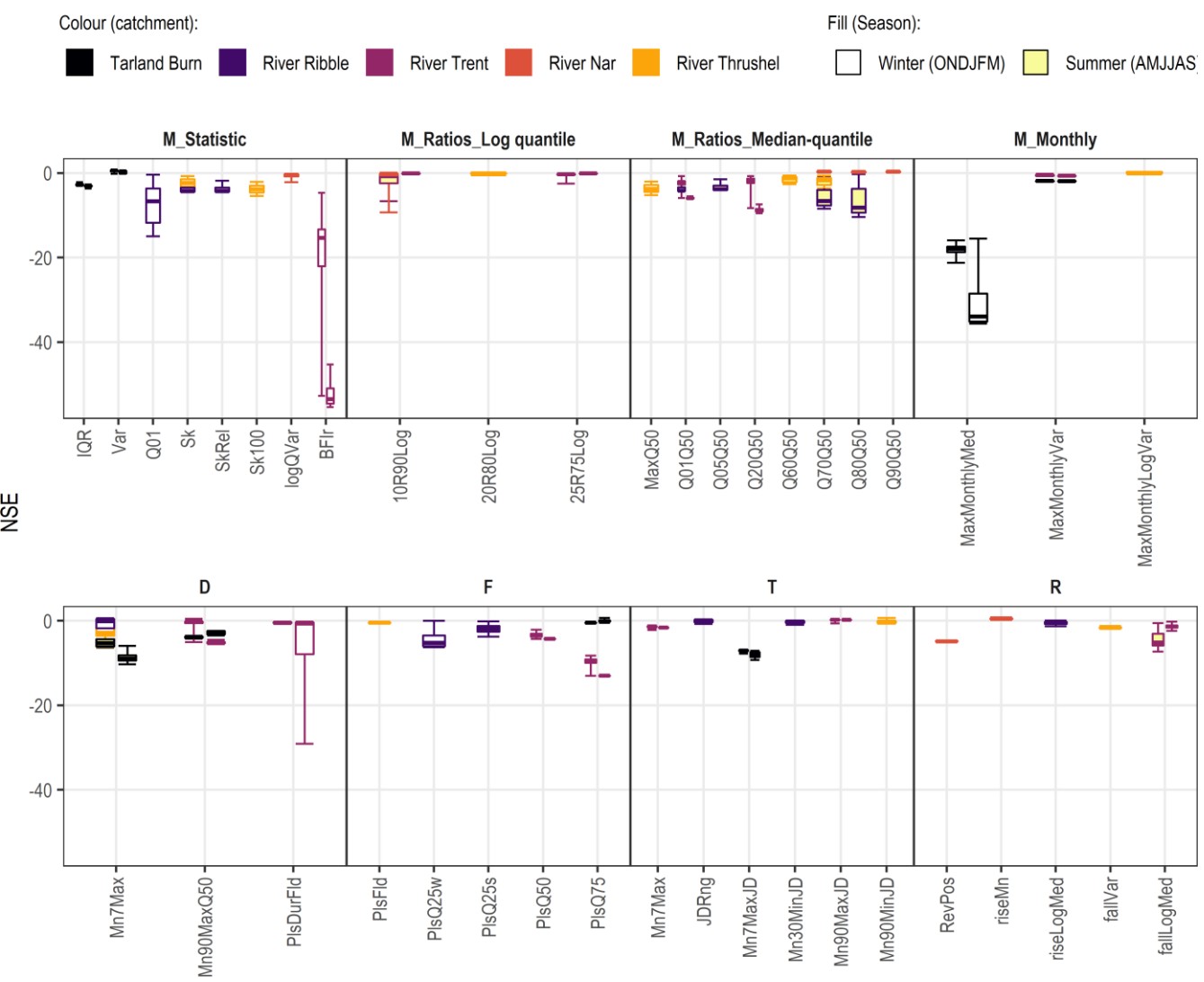

**Figure B2. Nash Sutcliffe Efficiency (NSE) for each ER HI; see Fig. 5 for NSE > 0. The 4- and 5-parameter results are adjacent, left and right respectively, for the Tarland Burn and River Trent.**

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
