# Peer review of "Replication of ecologically relevant hydrological indicators following a modified covariance approach to hydrological model parameterisation"

_Hydrology and Earth System Sciences, 2018_

## Short Comment (SC1) · 27 Nov 2018

Thanks for asking me for feedback for this interesting study on Researchgate. In general, I like this approach and think your work makes a good contribution towards better modelling of aspects in the hydrograph, which are relevant to hydroecology.

My main concern with this study is that you used only one catchment. In our studies on the subject (Vis et al., 2015; Pool et al., 2017, as cited in your manuscript) we used 25 catchments and actually found that the performances differed among catchments. This means that there is a risk for somewhat random results if one uses only one catchment and more catchments would be advisable for robust results. At the very least this
needs to be discussed, and it would be even better to extend your study to a few more catchments. Of course, handling all the simulations and their results can be painful as Marc (Vis), and Sandra (Pool) certainly will confirm.

Isn't the covariance approach by Vogel and Sankrarasubramanian (2003, WRR) not also some form of calibration /fitting. Could you clarify the difference to traditional calibration a bit more?

In the results, you report model parameters (P8 L7ff). Here it seems you derived one set of values. Later (P14) you discuss equifinality, but it was not fully clear to me how you considered equifinality in your study. Also, I am not sure I understand your comment regarding the number of model runs (P14L15): why would your 100 000 runs scale to 400 000 runs for the HBV model? It seems like you are arguing with the number of parameters, 4 to 16 resulting in a factor 4. However, this should not be a factor, but the exponent resulting in a far larger increase of model runs to sample the parameter space with a similar density.

In the discussion (P11L14) you say that the replication of indices was good. Does this result apply for calibration or validation (both regarding time period but even more regarding index)? In our studies, we found in general the indices could be nicely (even perfectly) be replicated when calibrated, but performances decreased for the validation case.

---

## Referee Comment (RC1) · Anonymous Referee #1 · 11 Dec 2018

**Review of the manuscript "Replication of ecologically relevant hydrological indicators following a covariance approach to hydrological model parameterisation" by Visser et al.**

In this manuscript, Visser et al. evaluate the ability of the hydrological model GR4J to reproduce multiple hydrological indicators in a study catchment in south-eastern England. In a first step, they create a random set of 100'000 parameter sets from which, in a second step, they select a single parameter set that reproduces various efficiency metrics within acceptable limits. The efficiency metrics include the covariance

between precipitation and streamflow, the covariance between potential evaporation and streamflow, and seven hydrological indicators that were found to be important in the study area. The acceptable limits are defined for each efficiency criteria separately based on the relative importance of a particular criteria in the study catchment. Model simulations from the best parameter set are evaluated by calculating various error metrics for each of the seven hydrological indicators. Finally, results are compared to existing studies by means of discussion.

I like the research questions of this manuscript and I see their relevance for practical implementations in the study area and current research in the domain of ecological flows. I especially like the novel idea of defining variable limits of acceptability based on the importance of efficiency criteria and I think that this is an approach worth to be analysed in more detail. I see the current level of the manuscript as an interesting starting point for a much more extended analysis or an in-depth analysis. Overall, I think the manuscript would benefit from a less generic introduction, which provides the reader a tailored and up to date background on modelling hydrological indicators and therefore sets the foundation for the final research questions. I also think that the research questions themselves should be addressed in more detail to support the final conclusions. The current study set up is very site-specific and it would be interesting to extend the analysis to more catchments to be able to generate more generic conclusions.

I hope that the comments below will be helpful for the authors to improve their manuscript.

**General comments**
The study aims at evaluating three research questions, which I think are very interesting. However, I have some concerns about the way the research questions are

addressed:

1. Research question 1 is addressed by using one study catchment, one hydrological model, and one parameter set. To make more general conclusions I highly recommend to address the question by using many more catchments or multiple hydrological models. Given the current equifinality-paradigm in hydrology, I also recommend to account for uncertainty by using many parameter sets.

2. Research question 2 about the comparison of various modelling studies is addressed by means of discussion. I am not sure if a discussion is enough to answer a research question. To me a research question should, if possible, be addressed by an analysis. If you wish to keep the comparison of your results with prior studies as a research question, I recommend to compare the studies in a quantitative way. Would it be possible that you contact the authors of the four studies to get access to more information?

3. Research question 3 is again addressed by means of discussion without any explicit analysis. The goal of question 3 is to address the limitations of classical calibration such as i) effect of data uncertainty, ii) effect of thresholds applied to select behavioural parameter sets, and iii) effect of equifinality. I wonder if the current study set up allows to tackle these challenges. For example, it would be important that you could show that your proposed approach is less sensitive to disinformative data than other approaches. Or it would be helpful if you could show/ discuss in more detail how the selection of a threshold (limits of acceptability) in this study is different from other studies. And finally, it would be interesting to see how the proposed covariance approach reduces equifinality compared to other approaches.

**Specific comments**
1. The authors motivate their research by stating that their approach is a step away from classical calibration. However, I don't understand in which way the proposed approach is different from calibration. I agree that finding acceptable parameter sets by some kind of optimization algorithm is different from finding acceptable parameter sets by a Monte Carlo approach. However, the approaches only differ in the way parameter sets are generated, whereby both approaches require at some stage the selection of parameter sets by means of efficiency criteria. To me, both approaches can therefore be considered as model calibration. A non-calibrated model to me would be one where the 100'000 randomly generated parameter sets are used without making any further selection. To me it would be important that you come up with convincing arguments for the statement that the proposed covariance approach is not a (multi-objective) model calibration.

2. The topic of multi-objective calibration leads me to my next comment. It is multiple times mentioned (e.g. title or research question 1) that a covariance approach was used to determine the most suitable model parameters. If I understand correctly, the final selection of a parameter set is based on the combined evaluation of the covariance between precipitation and streamflow, the covariance between potential evaporation and streamflow, and seven hydrological indicators. I would therefore argue that it is not a pure covariance approach, but rather a multi-objective approach that includes covariance as one out of multiple efficiency criteria. Additionally, I think that covariance can be considered as a classical signature with the novelty that it is not a pure hydrological signature, but rather a hydroclimatic signature. To me, the very interesting part is the fact that the objectives (efficiency criteria) used in this multi-objective function are weighted by their importance. Concluding, I would recommend to replace the term "covariance approach" by a term such as "multi-signature approach" or "multi-objective approach".

3. Is it correct that you select the final parameter set using the information of all 54

years? If yes, this would mean that you use the complete time series to find a parameter set and that there is no validation time period (meaning that all the error metrics are calculated for the calibration period). Since you have such a long time series, I would recommend to split the time series and use one part for an independent validation of the proposed approach.

4. The "calibration" finally leads to the selection of a single parameter set. Why do you use only one parameter set? Is it because there was only one out of the 100'000 parameter sets that was behavioural? If there are multiple behavioural parameter sets I strongly recommend to use all of them. Otherwise, you will need very good arguments for putting all your confidence on a single model output.

5. As far as I understood, hydrological indicators were calculated for each single year. Given that hydrological indicators were shown to be only robust if calculated over many years, how do you think this influences your results? Do you think that the yearly variability of the indicators can obscure/influence the uncertainty coming from the approach?

6. I think it would be worth to spend some time in adapting the introduction. For example, the two first paragraphs are very generic and I am not sure how much information they contain related to your research questions. You could shorten these paragraphs to one/two sentences and then extend the introduction to provide more background on e.g. multi-objective calibration or other studies modelling hydrological indicators.

**Detailed comments**

1. Abstract: You mention in the abstract that one benefit of the proposed approach is the reduction in overall time-demands. Could you specify what exactly you are

thinking of? The first thing that comes into my mind is that you want to reduce computational time. However, GR4J is a model, which is not very demanding in terms of computational effort. I was therefore wondering if you thought about using a more time-consuming model to proof that the approach does save computational time. This would also need a comparison of a traditional approach to your approach, which, I know, will need quite some time. Of course, you could also just explain or weaken your statement.

2. Study area: You mention that the catchment is an SSSI, that there is significant pressure on the river, and that the river has a highly seasonal flow regime. I think it would be interesting to add a sentence or two saying why the catchment is an SSSI, what kind of pressure sources exist, and how the seasonality looks like (mostly winter streamflow?).

3. Fig. 1: The markers for River Nar and Lexham village are difficult to differentiate in the figure.

4. Fig. 2: I was wondering why you decided to show a Figure of the model structure of GR4J? Given that you don't compare multiple models with different structures or that you don't extensively discuss model parameter values, I think you could remove the figure.

5. Table 2: I suggest to name the last item of the header "relative importance".

6. P6 L18: If I am not wrong, the reference to Fig. A2 comes before the reference to Fig. A1. So maybe you could switch the position of these two figures in the appendix.

7. P7 L4: Could you say specifically which error you minimise between observed and simulated covariance and HI? Is it the percent error?

8. Fig. 4: 1) The y-axis label is "percent error" while the legend says "difference between observed and simulated values" – what is correct? 2) The plot location of the hydrological indicators on the x-axis does not fully agree with the values in Table 2, e.g. Q70Q50 has according to Table 2 a low relative importance while it has a high one in the figure. 3) You use this figure two illustrate the concept of the limits of acceptability and to show the result of the best parameter set. Is there a way you could separate methods and results part in this figure?

9. P8 L6: The first reference you do in the results part is to a figure in the appendix (Fig. A2). Given the importance of this figure, I would suggest to have it in the main body of the manuscript.

10. P9 L7: You mention that you evaluate the model in terms of performance and consistency. I would therefore recommend that you rearrange the results chapter do this evaluation in a very clear way.

11. P9 L25-28: I would delete the first two sentences of this paragraph because they are methods and not results. I would also delete the last sentence of this paragraph and add the reference to Fig. 8 somewhere in brackets.

12. Figure captions: Could you be more specific in the figure captions, i.e. could you for each figure say how many data points are in each plot? I think it is important to guide the reader by telling if e.g. a histogram contains 54 simulation years or n parameter sets.

13. Fig. 5 and Fig. 6: These threes (sub)plots contain very similar information. I recommend to find a way to condense the information into a single figure. In Fig. 5a, what do the numbers in the brackets of the header mean?

14. Fig. 6: The figures contain a relatively small number of points. I was wondering if you can merge the three figures or if a table/ heatmap would be more suitable to show the results?

15. Fig. 8: The figure does not contain a dot for the 0-25 quantile of RevPos. I would suggest that you mention the reason for that in the figure caption and not somewhere in the main text. Maybe you could also plot the dot at the margin of the figure together with an arrow indicating that it is an outlier.

16. P21 L11: The reference to Cramer is not at the correct location and is lacking a year.

―――――――――――――――――

---

## Short Comment (SC2) · 7 Jan 2019

Hi and many thanks for your comments and feedback on this draft of the manuscript. I am currently in the process of updating in line with the suggestions of yourself (and reviewer 1), and aim to upload a revised draft in the next week.

---

## Short Comment (SC3) · 7 Jan 2019

Hi and many thanks for your detailed comments and feedback on this draft of the manuscript. I am currently in the process of updating in line with the suggestions of yourself (and Jan Seibert), and aim to upload a revised draft in the next week.

---

## Referee Comment (RC2) · Anonymous Referee #2 · 10 Jan 2019

This manuscript presents a modified approach after Vogel and Sankarasubramanian's covariance approach and extend the scope from a single-variable problem to a multiple-variable problem. The authors found that the approach can reduce model uncertainty and also time consumption. Overall, I think this is a great idea and the proposed approach and the results are a valuable contribution to the hydrological community. I recommend its publication after the following comments are addressed.

General comments:

1. The work is very site-specific and model-specific, which needs to be tackled or at least acknowledged.
2. Research Question 1 is addressed in the manuscript with direct, quantitative analysis but Questions 2 3 are not. My recommendation to the authors is to (1) provide more in-depth analysis for these two questions or (2) modify their research questions in Introduction. I feel that Question 1 can be listed as the single research question of this work, whereas your questions 2 and 3 can be raised in the Discussion section.

3. I think it helps the readers a lot if the authors can provide some clarification on how the 54 years of data were used in their approach. Was there any split? If so, which part is used as calibration and which part for validation? Why were HIs calculated for each year?

4. There can be some relocation of the figures and tables. For example, I think Figure 2 and Table 1 can be moved to Appendix or Supporting Information, since the hydrological model is already published and it is not the goal of this work to investigate the model itself. To the contrary, some of the Appendix information is critical and should be placed in the main text, e.g., Figure A1, A2, Table B1.

Specific comments:

5. P1L9: Does Vogel and Sankarasubramanian's covariance approach need a citation in the abstract? At least the journal and year of that publication should be provided.

6. P1 Introduction: The first two paragraphs can be largely shortened. Some of the details are well known and may not be necessary.

7. P2L30: Can you be more specific on "many of these problems"?

8. P3L29: By definition, Q90 should be larger than Q10. So Q95 below should be Q5.

9. P5: The "IT approach" might be presented with more details.

10. P6: Figure 3 may be modified by highlighting the boxes or flows that represent your new approach, as compared with the original 2003 approach.

11. P6L16: Check for consistency in the tense of verbs.

12. P6L18: Fix "the of the"

13. P7: Figure 4, A dashed horizonal line at 0.0 may be added.

14. P8: Table 3, it is not readily clear to me what "p>0.05" represents.

15. P9: Some of the context in the Results section should belong to Methods, e.g., the description of HAF.

16. P9L21: "précised"?

17. P11: Note the error "increas" in Figure 8 caption.

18. P11: The first three subsections of Section may contain a bracket noting the correspondence to the Research Questions. 19. P19: Note the error in Table B2 caption.

---

## Author Comment (AC2) · 22 Jan 2019

Dear editor, reviewers and Jan Seibert,

We would like to thank the reviewers and Jan Seibert for time they spent in providing such detailed reviews for our manuscript. A common theme amongst the comments was a need to consider more cases. To this end, please first see the general responses below.

**General responses**

**GR1 – Case-study catchments**

The availability and co-location of long-term hydroecological datasets represents the principle limiting factor in the application of the proposed modified covariance approach. This has been highlighted in both the methods and discussion sections of the revised manuscript.

A hydroecological dataset had recently been provided by the James Hutton Institute for a Scottish case study. Further, a scoping study of the English BIOSYS dataset and available hydroclimatological identified saw the identification of three further hydrologically diverse catchments. Additional factors we considered when identifying additional case studies were land use, BFI, location (north to south), seasonality and varied lengths of time-series. Thus, the consistency of the approach across a range of catchments is illustrated.

In addition to facilitating more general conclusions, this change allowed for the consideration of 40 distinct ER HIs covering all five facets of the flow regime. This represents a notable advantage over previous studies (Shrestha, Vis and Pool) which have been less diverse, focussing on a single suite of ER HIs and sub-basins of a larger catchment.

**GR2 – Hydrological models**

With respect to the number of hydrological models considered, the initial work on the River Nar did consider GR5J and GR6J. However, in stage 2, the models were invalidated. In this revision, this information has been included in the manuscript. Further, two out of the five catchments saw the validation of a small number of GR5J parameter sets.

Any further consideration of alternative model structures is deemed unnecessary (not commonplace in studies of a similar nature and beyond the scope of this work).

**GR3 – Time-series length, parameter sets and equifinality**

The validation of the model structure is achieved through comparison of the observed and simulated moments for covariance structure and ER HIs. This utilises the full length of the time-series. If observed and simulated moments are in agreement, the model is validated. To reduce the n parameter sets to a plausible parameter space, the log-threshold (limits of acceptability) is determined. For the parameter sets lying with this space, the time-series of annual ER HIs is subject to evaluation. In the original manuscript, a single parameter set was considered. In this revision, approximately 20 parameter sets per catchment (100 in total) are considered, serving to better highlight the performance and consistency across parameter sets.

The above revisions to the methodological application are also detailed in a figure at the end of this document. To reflect the changes in application, the introduction has been reworked to provide an overview of the development of methodological

approaches. Additionally, the research questions have been refocused to a single aim, evaluating the performance of the modified covariance approach. We hope that these changes address your most pressing concerns and that you will reconsider this revised manuscript for publication.

Responses on a comment-by-comment basis are provided below (alternatively see attachment 2 for excel format), followed by the revised manuscript. Please note that the revised manuscript does not have tracked changes. These were removed due to the extent of the required changes (moving sections/figures, deletions and additions). If this is still required, we would be happy to provide a document with tracked changes.

Yours sincerely,

Annie Visser (Corresponding author)

[Figure]

**Response to comments**

**Short comment – Jan Seibert**

| No. | (1) Comments from referees/public | (2) Author's response | (3) Author's changes in manuscript |
|---|---|---|---|
| 1 | Thanks for asking me for feedback for this interesting study on Researchgate. In general, I like this approach and think your work makes a good contribution towards better modelling of aspects in the hydrograph, which are relevant to hydroecology. | We thank you for your positive words towards the general concept of the presented approach. | - |
| 2 | My main concern with this study is that you used only one catchment. In our studies on the subject (Vis et al., 2015; Pool et al., 2017, as cited in your manuscript) we used 25 catchments and actually found that the performances differed among catchments. This means that there is a risk for somewhat random results if one uses only one catchment and more catchments would be advisable for robust results.
At the very least this needs to be discussed, and it would be even better to extend your study to a few more catchments. Of course, handling all the simulations and their results can be painful as Marc (Vis), and Sandra (Pool) certainly will confirm. | Thank you for your suggestion. We agree and obtained the necessary hydroecological data to allow for the application of the methodology to four additional hydrologically diverse catchments. | See figure and general responses. |
| 3 | Isn't the covariance approach by Vogel and Sankrarasubramanian (2003, WRR) not also some form of calibration/fitting. Could you clarify the difference to traditional calibration a bit more? | The motivation for the consideration of covariances is to allow the hydrological model to be validated (or invalidated) prior to any attempt at parameterisation. Additionally, the approach allows for the identification of a plausible parameter space. | The closing paragraph of the introduction has been reworked to provide a better overview of the approach by Vogel and Sankarasubramanian. Further, the methodology has been more clearly delineated to improve clarity of the steps required. We hope these changes meet with your approval. |
| 4 | In the results, you report model parameters (P8 L7ff). Here it seems you derived one set of values. Later (P14) you discuss equifinality, but it was not fully clear to me how you considered equifinality in your study. | In the original manuscript we looked to the entire parameter space to validate the model. It was also indicated, though not applied, that additional parameter sets may be easily considered as all simulations/HEs are | The results from the a plausible parameter space which encompasses ~20 parameter sets per catchment are presented in this revised manuscript. An attempt to clarify the discussion of equifinality in the text has also |

| | | | |
|---|---|---|---|
| | | considered in the application of the methodology. | been made (now in section 4.2.1).

Additionally, see figure and general responses. |
| 5 | Also, I am not sure I understand your comment regarding the number of model runs (P14L15): why would your 100 000 runs scale to 400 000 runs for the HBV model? It seems like you are arguing with the number of parameters, 4 to 16 resulting in a factor 4. However, this should not be a factor, but the exponent resulting in a far larger increase of model runs to sample the parameter space with a similar density. | Thank you for this clarification. However, a subsequent sensitivity analysis using Sobol-sequencing highlighted that the number of parameter sets considered for the initial case study (River Nar, n = 100,000) was excessive. Subsequently, we were able to successfully reduce the number of parameter sets to a more manageable 10,000 without any loss of information. | This was no longer considered relevant, given the addition of four additional catchments, and has thus been removed in the revised manuscript for the sake of conciseness. |
| 6 | In the discussion (P11L14) you say that the replication of indices was good. Does this result apply for calibration or validation (both regarding time period but even more regarding index)? In our studies, we found in general the indices could be nicely (even perfectly) be replicated when calibrated, but performances decreased for the validation case. | We recognise that the data used in validation and subsequent parameterisation was not clear in the original manuscript. | This has been clarified in Section 2.4 (stages 1 and 2). We hope this change provides the necessary clarification. |

**Anonymous reviewer 1**

| No. | (1) Comments from referees/public | (2) Author's response | (3) Author's changes in manuscript |
|---|---|---|---|
| 1 | In this manuscript, Visser et al. evaluate the ability of the hydrological model GR4J to reproduce multiple hydrological indicators in a study catchment in south-eastern England.

… etc. … | We thank you for your detailed and extensive review of the manuscript, it is very much appreciated. We hope that this revised manuscript addresses your concerns (specifically the 3 general comments). | See cover letter. |

| | | | |
|---|---|---|---|
| 2 | **General comments**
The study aims at evaluating three research questions, which I think are very interesting. However, I have some concerns about the way the research questions are addressed:

1. Research question 1 is addressed by using one study catchment, one hydrological model, and one parameter set. To make more general conclusions I highly recommend to address the question by using many more catchments or multiple hydrological models. Given the current equifinality-paradigm in hydrology, I also recommend to account for uncertainty by using many parameter sets. | Thank you for your suggestions. We agree with the need for additional study catchments and obtained the necessary hydroecological data to allow for the application of the methodology to four additional hydrologically diverse catchments.

We have now included three hydrological models across five catchments, n = 10,000 parameter sets for each, giving ~20 potential models per catchment. | We have made every effort to address your concerns through expanding the application and providing greater clarification in the text.

See figure and general responses (cover letter) for overview. |
| 3 | 2. Research question 2 about the comparison of various modelling studies is addressed by means of discussion. I am not sure if a discussion is enough to answer a research question. To me a research question should, if possible, be addressed by an analysis. If you wish to keep the comparison of your results with prior studies as a research question, I recommend to compare the studies in a quantitative way. Would it be possible that you contact the authors of the four studies to get access to more information? | - | In light of the feedback from reviewer 2, we have refocused the paper to consider a single aim. Thus, this research question has been removed; though some text remains in the discussion. |

| | | | |
|---|---|---|---|
| 4 | 3. Research question 3 is again addressed by means of discussion without any explicit analysis. The goal of question 3 is to address the limitations of classical calibration such as i) effect of data uncertainty, ii) effect of thresholds applied to select behavioural parameter sets, and iii) effect of equifinality. I wonder if the current study set up allows to tackle these challenges. For example, it would be important that you could show that your proposed approach is less sensitive to disinformative data than other approaches. Or it would be helpful if you could show/ discuss in more detail how the selection of a threshold (limits of acceptability) in this study is different from other studies. And finally, it would be interesting to see how the proposed covariance approach reduces equifinality compared to other approaches. | Whilst this is no longer a research question, some text remains in the discussion. See column (3).

With regards to disinformative data, we provide clear references to other work which has made these statements; therefore, it is our opinion that it is not necessary to consider this further.

The use of limits of acceptability is inherently different to traditional approaches as there is no reliance on an optimisation algorithm and arbitrary objective function. Instead, a plausible parameter space is determined through explicit consideration of the statistical importance of each hydrological indicator.

The aim of the approach is not to reduce equifinality, but rather to aid in the identification of a plausible parameter space. We recognise this was not clear in the original manuscript and hope that the revisions provide greater clarification. | In light of the feedback from reviewer 2, we have refocused the paper to consider a single aim. Thus, this research question has been removed; though some text remains in the discussion. |
| 5 | Specific comments
1. The authors motivate their research by stating that their approach is a step away from classical calibration. However, I don't understand in which way the proposed approach is different from calibration. I agree that finding acceptable parameter sets by some kind of optimization algorithm is different from finding acceptable parameter sets by a Monte Carlo approach. However, the approaches only differ in the way parameter sets are generated, whereby both approaches require at some stage the selection of parameter sets by means of efficiency criteria. To me, both approaches can therefore be considered as model calibration. A noncalibrated model to me would be one where the 100'000 randomly generated parameter sets are used without making any further selection. To me it would be important that you come up with convincing arguments for the statement that the proposed covariance approach is not a (multi-objective) model calibration. | In the manuscript we state that the approach is different to traditional calibration which relies on optimisation algorithms and objective functions. The modified covariance approach does not use any optimisation algorithm, instead the entire parameter space is simulated and ER HIs derived. From this, the parameter sets are reduced to a plausible parameter range based on the statistical importance of the ER HIs. Additionally, validation is performed prior to parameterisation of the model. | We have rewritten the discussion of the approach in the final paragraph of the introduction and attempt to clarify section 2.4.1 in the methods.

The reworking of the introduction more generally should also serve to place the approach in the wider context. |

| | | | |
|---|---|---|---|
| 6 | 2. The topic of multi-objective calibration leads me to my next comment. It is multiple times mentioned (e.g. title or research question 1) that a covariance approach was used to determine the most suitable model parameters. If I understand correctly, the final selection of a parameter set is based on the combined evaluation of the covariance between precipitation and streamflow, the covariance between potential evaporation and streamflow, and seven hydrological indicators. I would therefore argue that it is not a pure covariance approach, but rather a multi-objective approach that includes covariance as one out of multiple efficiency criteria. Additionally, I think that covariance can be considered as a classical signature with the novelty that it is not a pure hydrological signature, but rather a hydroclimatic signature. To me, the very interesting part is the fact that the objectives (efficiency criteria) used in this multi-objective function are weighted by their importance. Concluding, I would recommend to replace the term "covariance approach" by a term such as "multi-signature approach" or "multi-objective approach". | The method has been named to reflect Vogel and Sankarasubramanian (2003) where they referred to their method as the 'covariance approach'. Our modifications sees the consideration of hydroecological modelling outcomes through the limits of acceptability.

Further, we feel that the method is at odds with the descriptions of multi-criteria approaches in other works. For instance, in their review of multi-criteria approaches, Efstratiadis and Koutsoyiannis (2010; https://doi.org/10.1080/02626660903526292) describe the use of algorithms and search techniques. To avoid confusion, and ensure consistency with the original method, we feel it is more appropriate to retain the title as is. Additionally, other studies focussing on replicating aspects of the time-series are not termed multi-criteria, e.g. Westerberg et al. (2011; https://doi.org/10.5194/hess-15-2205-2011).

Additionally, it is worth highlighting that we have referred to this (in brief) as a modified covariance approach in the keystone paper of this project which is published in EMS: https://doi.org/10.1016/j.envsoft.2019.01.004 | No change. Title has been updated to say "modified covariance approach" - this was an oversight on our part. |
| 7 | 3. Is it correct that you select the final parameter set using the information of all 54 years? If yes, this would mean that you use the complete time series to find a parameter set and that there is no validation time period (meaning that all the error metrics are calculated for the calibration period). Since you have such a long time series, I would recommend to split the time series and use one part for an independent validation of the proposed approach. | We agree that split-sampling is the traditional approach - this was acknowledged in the original manuscript. However, one of the advantages of this approach is that validation and identification of a plausible parameter space is determined through covariances, thereby avoiding the need for splitting the time-series. Additionally, the other case studies do not have time-series as long as this (a necessary consequence of identifying catchments where sufficient hydroecological data is available). | Section 2.4 in the methods has been updated to clarify this point. |

| | | | |
|---|---|---|---|
| 8 | 4. The "calibration" finally leads to the selection of a single parameter set. Why do you use only one parameter set? Is it because there was only one out of the 100'000 parameter sets that was behavioural? If there are multiple behavioural parameter sets I strongly recommend to use all of them. Otherwise, you will need very good arguments for putting all your confidence on a single model output. | The moments in the appendix (now figure A3) indicate excellent agreement between the observations and simulations. Indeed, if only a single parameter set was deemed suitable then it may call into question the validity of the hydrological model.

In the original manuscript the validation did consider all 100,000. However, these were not all subsequently analysed in the evaluation. | In the updated manuscript we use the limits of acceptability to identify a plausible parameter space of ~20 parameter sets that minimises the covariance error (this is not fixed - it is simply for illustrative purposes). Consideration of these 20 parameter sets further allows for expanded discussion of consistency. |
| 9 | 5. As far as I understood, hydrological indicators were calculated for each single year. Given that hydrological indicators were shown to be only robust if calculated over many years, how do you think this influences your results? Do you think that the yearly variability of the indicators can obscure/influence the uncertainty coming from the approach? | Validation and identification of the feasible parameter space looks at the full length of the time-series. Model evaluation looks at annual indicators as their projection represents the overall modelling objective. | An additional comment has been added to the close of the paragraph in section 4.2 Advantages and limitations of the modified covariance approach. |
| 10 | 6. I think it would be worth to spend some time in adapting the introduction. For example, the two first paragraphs are very generic and I am not sure how much information they contain related to your research questions. You could shorten these paragraphs to one/two sentences and then extend the introduction to provide more background on e.g. multi-objective calibration or other studies modelling hydrological indicators. | - | We appreciate this suggestion and have adapted accordingly. The introduction is now more focussed on background and the development of methods over time. |

| | Detailed comments | | |
|---|---|---|---|
| 11 | 1. Abstract: You mention in the abstract that one benefit of the proposed approach is the reduction in overall time-demands. Could you specify what exactly you are thinking of? The first thing that comes into my mind is that you want to reduce computational time. However, GR4J is a model, which is not very demanding in terms of computational effort. I was therefore wondering if you thought about using a more time-consuming model to proof that the approach does save computational time. This would also need a comparison of a traditional approach to your approach, which, I know, will need quite some time. Of course, you could also just explain or weaken your statement. | This was indicated in the original manuscript (in revised manuscript it is the last paragraph of section 4.2.1 General advantages). However, this does represent an interesting opportunity for future work - a comparison of hydrological models in terms of computational time, performance and consistency. | Removed from abstract. Only consideration now is in the discussion. |
| 12 | 2. Study area: You mention that the catchment is an SSSI, that there is significant pressure on the river, and that the river has a highly seasonal flow regime. I think it would be interesting to add a sentence or two saying why the catchment is an SSSI, what kind of pressure sources exist, and how the seasonality looks like (mostly winter streamflow?). | The seasonality is captured in the ER HIs. With the addition of four catchments, the level of detail on each catchment has been reduced to a level reflecting similar studies (summary table + map). | - |
| 13 | 3. Fig. 1: The markers for River Nar and Lexham village are difficult to differentiate in the figure. | This level of detail was no longer necessary, figure is thus redundant and has been removed. | - |
| 14 | 4. Fig. 2: I was wondering why you decided to show a Figure of the model structure of GR4J? Given that you don't compare multiple models with different structures or that you don't extensively discuss model parameter values, I think you could remove the figure. | - | This has been moved to the appendix. Description of the additional hydrological models has also been added. |
| 15 | 5. Table 2: I suggest to name the last item of the header "relative importance". | This was not relative importance as it had not been standardised to a range of zero to one. However, the Table has been removed as it was no longer relevant. | However, it was observed that in the text the term relative had been used incorrectly towards the beginning. This has been corrected for clarification. |
| 16 | 6. P6 L18: If I am not wrong, the reference to Fig. A2 comes before the reference to Fig. A1. So maybe you could switch the position of these two figures in the appendix. | - | This has been addressed in the revised manuscript. |

| | | | |
|---|---|---|---|
| 17 | 7. P7 L4: Could you say specifically which error you minimise between observed and simulated covariance and HI? Is it the percent error? | - | Text updated: "the ability to replicate or minimise the error (percentage difference)" |
| 18 | 8. Fig. 4:
 1) The y-axis label is "percent error" while the legend says "difference between observed and simulated values" – what is correct?
 2) The plot location of the hydrological indicators on the x-axis does not fully agree with the values in Table 2, e.g. Q70Q50 has according to Table 2 a low relative importance while it has a high one in the figure.
 3) You use this figure two illustrate the concept of the limits of acceptability and to show the result of the best parameter set. Is there a way you could separate methods and results part in this figure? | 1) No longer relevant and clarified in #17.
 2) No longer relevant. Error was the result of mislabelling of the points in the figure.
 3) See column (3). | The figure has been adjusted to be a simple exemplar which does not use real data. |
| 19 | 9. P8 L6: The first reference you do in the results part is to a figure in the appendix (Fig. A2). Given the importance of this figure, I would suggest to have it in the main body of the manuscript. | - | This figure is now only considered as an exemplar of how the moments may be visualised (methodology). |
| 20 | 10. P9 L7: You mention that you evaluate the model in terms of performance and consistency. I would therefore recommend that you rearrange the results chapter do this evaluation in a very clear way. | The terms performance and consistency are introduced in order to provide clear definition of terms. In the original manuscript, consistency was with respect to performance across the seven ER HIs. We appreciate your suggestion in that context. | In this revised manuscript, we look for consistency across parameter sets and catchments as well. With a view to minimising the length of the results section, results are presented in sub-sections relating to each of the tests applied. The tests are considered in the order they are applied, thus the addition of each test serves to advance the 'story'. |
| 21 | 11. P9 L25-28: I would delete the first two sentences of this paragraph because they are methods and not results. I would also delete the last sentence of this paragraph and add the reference to Fig. 8 somewhere in brackets. | Thank you for highlighting this. The latter is no longer relevant in the revised manuscript. | - |

| | | | |
|---|---|---|---|
| 22 | 12. Figure captions: Could you be more specific in the figure captions, i.e. could you for each figure say how many data points are in each plot? I think it is important to guide the reader by telling if e.g. a histogram contains 54 simulation years or n parameter sets. | - | In the revised manuscript 100 parameter sets are considered for 5 catchments (~20 per catchment). Section 2.5 model evaluation has been updated to clarify this point with an example for the River Nar included. This statement is true for all metrics applied. Results are presented as box plots to illustrate the range of values across the parameter sets. |
| 23 | 13. Fig. 5 and Fig. 6: These threes (sub)plots contain very similar information. I recommend to find a way to condense the information into a single figure. In Fig. 5a, what do the numbers in the brackets of the header mean? | - | In this revised manuscript the results from the statistical tests are presented in a tabular format. Given the size of the required table (with the increased number of catchments and ER HIs) this table is located in the appendix. |
| 24 | 14. Fig. 6: The figures contain a relatively small number of points. I was wondering if you can merge the three figures or if a table/ heatmap would be more suitable to show the results? | - | As above in #23. |
| 25 | 15. Fig. 8: The figure does not contain a dot for the 0-25 quantile of RevPos. I would suggest that you mention the reason for that in the figure caption and not somewhere in the main text. Maybe you could also plot the dot at the margin of the figure together with an arrow indicating that it is an outlier. | We thank you for this comment and will certainly bare this in mind for future work. In this instance, the consideration of additional catchments have seen this figure evolve considerably. | Not relevant. Figure replaced. |
| 26 | 16. P21 L11: The reference to Cramer is not at the correct location and is lacking a year. | - | To ensure consistency, all references which were located within Table 3 have been moved to the main body text. |

**Anonymous reviewer 2**

| No. | (1) Comments from referees/public | (2) Author's response | (3) Author's changes in manuscript |
|---|---|---|---|
| 1 | This manuscript presents a modified approach after Vogel and Sankarasubramanian's covariance approach and extend the scope from a single-variable problem to a multiple-variable problem. The authors found that the approach can reduce model uncertainty and also time consumption. Overall, I think this is a great idea and the proposed approach and the results are a valuable contribution to the hydrological community. I recommend its publication after the following comments are addressed. | We thank you for your feedback and suggestions. We hope the revisions meet with your approval. | Detailed below and see also cover letter. |
| 2 | General comments:
1. The work is very site-specific and model-specific, which needs to be tackled or at least acknowledged. | - | An additional four hydrologically diverse catchments have been considered (this is necessarily limited due to hydroecological data availability). We have expanded the paper to explicitly consider GR4J, 5J and 6J (these were considered in the original application but were invalidated). Any further consideration of alternative hydrological model structures is not deemed necessary as similar studies have not accounted for this. |
| 3 | 2. Research Question 1 is addressed in the manuscript with direct, quantitative analysis but Questions 2 3 are not. My recommendation to the authors is to (1) provide more in-depth analysis for these two questions or (2) modify their research questions in Introduction. I feel that Question 1 can be listed as the single research question of this work, whereas your questions 2 and 3 can be raised in the Discussion section. | - | We agree; additionally, the consideration of additional catchments and parameter sets allows for more generalised conclusions and therefore the second and third research questions are no longer necessary. Where relevant, the text in the discussion has been retained. |

| | | | |
|---|---|---|---|
| 4 | 3. I think it helps the readers a lot if the authors can provide some clarification on how the 54 years of data were used in their approach. Was there any split? If so, which part is used as calibration and which part for validation? Why were HIs calculated for each year? | Validation and identification of the feasible parameter space looks at the full length of the time-series. Model evaluation looks at annual indicators as their projection represents the overall modelling objective. | Clarification has been provided in section 2.4 Covariance approach with regards to the time-series.

An additional comment re annual indicator uncertainty has been added to the close of the paragraph in section 4.2 Advantages and limitations of the modified covariance approach. |
| 5 | 4. There can be some relocation of the figures and tables. For example, I think Figure 2 and Table 1 can be moved to Appendix or Supporting Information, since the hydrological model is already published and it is not the goal of this work to investigate the model itself. To the contrary, some of the Appendix information is critical and should be placed in the main text, e.g., Figure A1, A2, Table B1. | Figures and tables have been relocated.

Figure A1: No longer considered relevant due to the shift in focus of the paper. To account for this behaviour we adapted the HAF code for integer values.

Figure A2: This remains in the appendix as it now represents only an exemplar for one of the case studies (it is not practically possible to display these figures for all catchments and hydrological models, 5*3).

Table B1: With the adjustment of the research questions this table has been removed, the level of detail is no longer required. | - |
| 6 | Specific comments:
5. P1L9: Does Vogel and Sankarasubramanian's covariance approach need a citation in the abstract? At least the journal and year of that publication should be provided. | - | This modification has been made; it as the discretion of the editor/journal whether they will allow it (in my experience the removal of a citation has been requested). |
| 7 | 6. P1 Introduction: The first two paragraphs can be largely shortened. Some of the details are well known and may not be necessary. | - | The introduction has, generally, been revised to reflect the development of methods over time. |
| 8 | 7. P2L30: Can you be more specific on "many of these problems"? | - | This line has been deleted as the research questions have been modified. |

| | | | |
|---|---|---|---|
| 9 | 8. P3L29: By definition, Q90 should be larger than Q10. So Q95 below should be Q5. | We are aware that Q90 can mean low and high flows interchangeably. In the context of ER HIs, Q90 is the flow exceeded 90% of the time, hence, a low flow. See Shrestha et al., 2014, Vis et al., 2015 and Pool et al., 2017 for example. | - |
| 10 | 9. P5: The "IT approach" might be presented with more details. | - | Section 2.3 Data has been expanded to consider the hydroecological model development further. Additionally, further details on information theory are provided in the Appendix. |
| 11 | 10. P6: Figure 3 may be modified by highlighting the boxes or flows that represent your new approach, as compared with the original 2003 approach. | We agree this would be useful, however the level of difference is marginal. In stage 1 and 2 the only difference is that more than one HI is determined. The principle difference comes in the third stage and the identification of the plausible parameter space. | A new line has been added to the end of paragraph one in section 2.4 Covariance approach. |
| 12 | 11. P6L16: Check for consistency in the tense of verbs. | - | Addressed as appropriate. |
| 13 | 12. P6L18: Fix "the of the" | - | Addressed. |
| 14 | 13. P7: Figure 4, A dashed horizonal line at 0.0 may be added. | - | The revised figure based on an exemplar now depicts error. |
| 15 | 14. P8: Table 3, it is not readily clear to me what "p>0.05" represents. | - | Added clarification in text and caption that this refers to the significance 'threshold' specified. It has also been updated to 0.001. |
| 16 | 15. P9: Some of the context in the Results section should belong to Methods, e.g., the description of HAF | - | Addressed as appropriate. |
| 17 | 16. P9L21: "précised"? | Summarised. | Not present in revised text. |
| 18 | 17. P11: Note the error "increas" in Figure 8 caption. | - | Addressed. |
| 19 | 18. P11: The first three subsections of Section may contain a bracket noting the correspondence to the Research Questions. | - | Not relevant in the revised text. |
| 20 | 19. P19: Note the error in Table B2 caption. | | Not relevant in the revised text. |

[revised manuscript text omitted]

---

## Author Comment (AC3) · 22 Jan 2019

Thank you for your time in making these comments. Please see the attached supplement for (1) our general response, (2) response on a comment by comment basis and (3) the revised manuscript. Kind regards, Annie Visser

Please also note the supplement to this comment:
https://www.hydrol-earth-syst-sci-discuss.net/hess-2018-536/hess-2018-536-AC3-supplement.pdf

536, 2018.

---

## Referee Report (RR1)

**Review of the manuscript „Replication of ecologically relevant hydrological indicators following a modified covariance approach to hydrological model parameterisation" by Visser et al.**

Dear editor and authors,

I thank the authors for their well-structured answers to the comments. I especially appreciate that some major concerns have been addressed and are now improved in the current version of the manuscript. The use of 5 catchments instead of 1 catchment, the use of multiple hydrological models, and the use of 20 parameter sets instead of 1 parameter set strongly improved the quality of the work and clearly increased the relevance of the study for the ecohydrological community. The presentation of the results was also changed and results are now presented in a logically structured way. However, I still have some major concerns regarding some fundamental statements made in this manuscript about parameterization, objective functions, and split-sample tests.

Major comments

- It is multiple times highlighted that the proposed covariance approach has some considerable benefits over a traditional calibration approach. However, it is not clear what a traditional calibration approach is. A precise description of a traditional calibration approach is in my opinion highly important to be able to make a fair comparison between the two approaches.
- The covariance approach is described as an approach where model structure is validated (step 2 of approach) prior to parameterisation (step 3 of approach). I have two concerns regarding this statement. First, step 1 consists of sampling the parameter space and running simulations. So strictly speaking, simulations are run without validating the structure. To me a validation of the model structure without parameterization would e.g. be the check if the model structure represents the most important processes occurring in a catchment. For this no parameters are needed. Second, stage 2 of the approach is the visualization of the simulations in the form of efficiency plots, and stage 3 is the selection of parameter sets using covariance and ER His. Stage 2 is therefore not a validation – validation is only done in step 3 at the moment where a "judgment" is made about the validity of a parameter set.
- It is argued that the use of the covariance approach is a shift away from the use of objective functions. I don't agree with this statement. Objective functions are criteria used to objectively select behavioural parameter sets. In this study, covariance and ecologically relevant hydrological indicators (ER HI) are used to select behavioural parameter sets. So both the covariance and the ER His are objective functions. Given that parameter sets are selected based on a combination of covariance and ER HIs, I would argue that a "classical" multi-objective function is used in this study. (The authors referenced in their response to the comments the review of Efstratiadis and Koutsoyiannis, where it is acknowledged that the term multi-objective can refer to both a scalar of multiple criteria or a vector based optimization approach). I am still not so happy with the term covariance approach, because the selection of parameter sets is based on two covariance metrics and many more ER HIs. The approach is therefore named after one out of many metrics used.
- The statement that the proposed covariance approach "avoids the need for split-sampling" is highly critical. The idea of split-sampling is that one selects parameter sets in one time period and then tests these selected parameter sets in an independent (maybe climatically/

meteorologically different) time period. This safety-check is independent of the approach used to select the parameter sets.

- In the discussion (chap. 4.4) the covariance approach is suggested as a tool to be used in environmental flow assessment, climate change studies, water resources management, prediction in ungauged catchments, and model selection frameworks. I wonder if the approach is really a solution to many of the biggest challenges in hydrology?
- The results of this study are mostly compared against three other studies that model ER HIs (i.e., Pool et al., 2017; Vis et al., 2015, Shrestha et al., 2014). Could you maybe find more studies for comparison? There are quite a number of studies using hydrological signatures (not necessarily ecologically relevant) for model calibration or validation. Such studies could be helpful to discuss the covariance approach and the simulation results of this study.

Minor comments

- P18 L20 - Study areas: Could you maybe provide some information on how exactly you selected the 5 study catchments?
- P18 L28 and Table 1: To complete the information in Table 1 I would add the information about the altitude, especially since you explicitly mention that altitude is different between the 5 study catchments. I would also suggest to either provide the definition of BFI in the table caption or not to use the abbreviation. The * behind "principle land use" should probably also be added to the table caption. Finally, you could eventually add the river type to the table.
- P19 L8: You mention that Perrin et al. (2008), Coron et al. (2016), and others have been using the GRJ model series for a wide range of applications. To increase the information content of this sentence, I recommend to explicitly state what GRJ has been used for (e.g. GR4 was used for climate change modelling, the prediction in ungauged basins, etc.).
- P19 L10-20: Parameter x3 is missing in the description of the model parameters. Given that the focus of the study is not the development of a new model I wonder if such a detailed description is needed. I suggest keeping the model description more generic focusing on the runoff processes represented by the model.
- P20 chpt. 2.4: The covariance approach is closely related to the GLUE methodology proposed by Beven and Binley (1992). Both approaches generate in a first step a large set of parameters, from which a subset is selected using limits of acceptability. Where do you see the differences?
- P20 L14-16: To me it is not clear from the context of the paragraph for which model (hydroecological model or hydrological model) the described data is used. Also, references to data sources are missing.
- P21 Figure 1: The last column of stage 1 uses three boxes to show the model output. I would suggest to reduce it to two boxes. This could either be i) one box for covariance (observed and simulated) and one box for HI (observed and simulated) or ii) one box for simulations (covariance and HI) and one box for observation (covariance and HI).
- P22 L7-9: Could you shortly state why exactly you selected an e function and not a linear function? How did you decide that you want 20 parameter sets and not e.g. 30?
- P22 Fig. 2: The figure could be complemented by a legend to make it easier to understand. I would also suggest to use the grey area to highlight the plausible parameter space and not the parameter sets not used for the final simulations.
- P24 L6-8: The range of parameter values is used as an indicator for consistency. However, the possible/plausible values of the model parameters vary in order of magnitudes. E.g. x4 that is an indicator for the event length (days) has different plausible values than x1 that is an

indicator for the storage (mm). This leads to your result that those parameters having values in the order of 100 have a much lower consistency than those having values in the order of 1-10. I would either normalize the parameter values or not use the range as in indicator for consistency.

- P 27 L9: NSE is a measure where the hydrological model simulations are evaluated against the simple model of predicting mean discharge for every day.
- Section 3.2.5: In this section you compare the efficiencies across your study catchments and you don't present any new results. So I would move the whole section into the discussion chapter.
- Section 4.1: This section is a summary of your results chapter. In the discussion, to me it would be more interesting to read about why you have certain results. You actually do that in section 4.3.1. I would therefore merge section 4.1. and 4.3.1.
- P31 L1: I don't think that your statement "… practical limit to the number of ER HIs" is correct. Because if one e.g. uses an optimization algorithm with a multi-criteria objective function, then each criteria can have a certain weight (equal weights or as in your case weights according to importance). So I don't think that the number of criteria is limited from a practical perspective

Technical details

- P18 L23: What does WFD mean?
- P18 L23-25: I think the sentence "Under the modified covariance…." doesn't fit into the context of this paragraph and I would remove it.
- P20 L1: I would rename the chapter from "data" to e.g. "Selection of ecologically relevant hydrological indicators", because most information in this chapter is on the selection process of the ER HI.
- P20 L1-10: You provide a description of the hydroecological modelling approach in the appendix. I would therefore shorten this paragraph to the minimum information needed to understand how you selected the ER HIs.  Sentences such as "measure of the statistical weight…" or "Consequently, more conclusive statements may be made…" could be removed. However, I would complement the section with the information on how many ER HI were selected for each catchment, because this information is relevant for stage 3 in your modelling approach.
- P23 Table 3: I would extend the description of NSE: "…A measure of goodness of fit to the 1:1 line normalized by the variance". The detail on the normalization is important and you refer to in the discussion.
- P24 Table: Table 2 should be relabelled to table 4. I also suggest to add parameter x6 for completeness.
- Figures 3, 4, and 5: Please explain what the subplots titles, such as D, F, T, and R mean.
- I made the general observation that many Figures are not readable in black and white. Can you think of a way to change that?

---

## Referee Report (RR2)

**Review of the manuscript „Replication of ecologically relevant hydrological indicators following a modified covariance approach to hydrological model parameterisation" by Visser et al.**

Dear editor and authors,

I thank the authors for rigorously answering and addressing many of my comments. I highly appreciate their efforts. I only have a small number of comments left – some of them were already part of the last review round, but still need some clarification. Besides that, I would like to emphasise that I personally think that the manuscript is now of good quality and that the study proposes and tests an interesting approach to model parameterisation. It is definitively a valuable contribution for the (eco-) hydrological modelling community.

Comments

1)  P11 L 24-25: The introduction now includes a part that defines the "traditional" approach. However, the definition is still relatively vague, mostly because there is a lack of examples. Is the Nash-Sutcliffe efficiency a traditional objective function? What about the Kling-Gupta efficiency? What about the flow duration curve? These questions can be easily clarified by adding the information (e.g. "…traditional objective functions, such as the NSE and KGE…"). Also, it would be interesting to know which objective function was used in the studies that are referenced right after the definition (Shresta et al., 2014; Vis et al, 2015). This would certainly help to define what a traditional objective function is.

2)  P12 L10: You mention the selection of ER HIs to guide model parameterisation as one limitation you want to address in this study. The limits of acceptability concept of the covariance approach also needs a selection of ER HIs. Can you shortly comment on where the difference is?

3)  P12 L11 and P28 L27-32: Is the recalibration of a model, such as GRJ, really a limitation in terms of computational power? I assume the bigger challenge is rather to find a set of parameters that manages to replicate multiple ER HIs. Having such a "common" set of parameters is hydrologically more meaningful than having a separate set of parameters for different ER HIs.

4)  Table 1: Catchment steepness is missing for the Tarland Burn catchment. I assume that it could be calculated relatively easily using the catchment outline and a digital elevation model.

5)  This comment is on step 2 and 3 of the covariance approach. In Fig. 1 and the corresponding text you make a clear distinction between step 2 and step 3. I am not convinced that it is meaningful to separate the two steps. To my understanding you do the following: In step 2, you make a plot of the observed and $n$ simulated covariances and each ER HI. If the observed moments lie inside the "cloud" of simulated moments, then the model structure is valid. In step 3, you take a subset of the "cloud" that is within the limits of acceptability. This gives you the valid parameter sets. You therefore use a stricter test to validate parameter sets than to validate the model structure. My question is now: does that make sense? An extreme example: imagine you have 10,000 parameter sets, a validated model structure, but only 1 parameter set is within the limits of acceptability. Would you say you have a valid model structure if only 1 parameter set actually manages to give you simulations of covariance and ER HI that are realistic enough to work with? Personally, I would doubt that my model structure is valid. Based on your text and comments I understand that step 2 is exactly as proposed by Vogel and Sankrasubramanian and that step 3 is the extension. However, I think what you want to do is modifying their approach to 1) use it for multiple indicators, and 2) to weight indicators according to their importance. To do so, you can skip step 2 and directly apply step 3.

6) P28 L23: As you mention, equifinality is reaching the same outcome by different means. This is mainly due to uncertainties in data and model structure/parameters and the limited type of data used to evaluate the model. Given the same model and data, I don't fully understand why the covariance approach reduces equifinality compared to a traditional approach.

7) P30 L8: You mention that an improvement in consistency is reached by the covariance approach. Is this due to the covariance approach or due to the fact that all available ER HIs are used to select parameter sets, i.e. the ER HIs are part of the calibration?

8) Finally, I fully agree that your approach is different from the GLUE approach proposed by Beven and Binley (1992). However, they have a lot of similarity and this is why I think it is fair to shortly comment on that somewhere in the manuscript (two to three sentences are enough). I would like to add some thoughts to your previous answer to this topic: a) both, the GLUE and the covariance approach need a performance metric (you use the covariance and ER HIs; any objective function (also called likelihood function) can be used in GLUE), b) statistical importance could be considered in GLUE using the approach of fuzzy limits of acceptability (Beven, 2006), and c) a range of indicators can be considered in GLUE (e.g.Blazkova and Beven, 2009).

References

Beven, K. (2006). A manifesto for the equifinality thesis. *Journal of hydrology*, *320*(1-2), 18-36.

Blazkova, S., & Beven, K. (2009). A limits of acceptability approach to model evaluation and uncertainty estimation in flood frequency estimation by continuous simulation: Skalka catchment, Czech Republic. *Water Resources Research*, *45*(12).

---

## Author Response (AR2)

Dear Professor Pierre Gentine and reviewers,

We would like to thank the reviewers for their reconsideration of the manuscript, and the comments from reviewer 1. We would also like to thank the editor for advising where to focus our efforts in this revision. Broadly speaking, the comments were focussed on the use of terminology and how the results where put forth. We have taken these comments very seriously and have made changes based on thirty-one out of thirty-five comments. We hope that these changes are sufficient to address the concerns raised.

Responses on a comment-by-comment basis are provided below (alternatively see attachment 2 for excel format), followed by the revised manuscript. Comments number five, seven and eighteen required more extensive responses. These are provided below the table (alternatively see attachment 3). Track changes has been used in the manuscript – with the exception of the replacement of figures.

Yours sincerely,

Annie Visser (Corresponding author)

| Name | Number | Comment | Action taken | Response |
|---|---|---|---|---|
| colspan=5 | Comments 1 to 3 required no action and are therefore not shown here. | | | |
| Reviewer 1 | 4 | It is multiple times highlighted that the proposed covariance approach has some considerable benefits over a traditional calibration approach. However, it is not clear what a traditional calibration approach is. A precise description of a traditional calibration approach is in my opinion highly important to be able to make a fair comparison between the two approaches. | Yes | The introduction, and remaining text (as appropriate), has been revised to be consistent with the language in studies such as Westerberg et al. (2011) and Vogel and Sankarasubramanian (2003). The introductory paragraph where the 'traditional approach' is first introduced has been structured such that the example provides further explanation. |
| | 5 | The covariance approach is described as an approach where model structure is validated (step 2 of approach) prior to parameterisation (step 3 of approach). I have two concerns regarding this statement. First, step 1 consists of sampling the parameter space and running simulations. So strictly speaking, simulations are run without validating the structure. To me a validation of the model structure without parameterization would e.g. be the check if the model structure represents the most important processes occurring in a catchment. For this no parameters are needed. Second, stage 2 of the approach is the visualization of the simulations in the form of efficiency plots, and stage 3 is the selection of parameter sets using covariance and ER His. Stage 2 is therefore not a validation – validation is only done in step 3 at the moment where a "judgment" is made about the validity of a parameter set. | Yes | See detailed responses. |
| | 6 | It is argued that the use of the covariance approach is a shift away from the use of objective functions. I don't agree with this statement. Objective functions are criteria used to objectively select behavioural parameter sets. In this study, covariance and ecologically relevant hydrological indicators (ER HI) are used to select behavioural parameter sets. So both the covariance and the ER His are objective functions. Given that parameter sets are selected based on a combination of covariance and ER HIs, I would argue that a "classical" multi-objective function is used in this study. (The authors referenced in their response to | Yes | We appreciate the reviewer's comment. We feel that this has been addressed in the process of making revisions based on other comments; this statement, or similar, is no longer made in the text. |

| | | | |
|---|---|---|---|
| | the comments the review of Efstratiadis and Koutsoyiannis, where it is acknowleged that the term multi-objective can refer to both a scalar of multiple criteria or a vector based optimization approach). | | |
| 7 | I am still not so happy with the term covariance approach, because the selection of parameter sets is based on two covariance metrics and many more ER HIs. The approach is therefore named after one out of many metrics used. | No | See detailed responses. |
| 8 | The statement that the proposed covariance approach "avoids the need for split-sampling" is highly critical. The idea of split-sampling is that one selects parameter sets in one time period and then tests these selected parameter sets in an independent (maybe climatically/meteorologically different) time period. This safety-check is independent of the approach used to select the parameter sets | Yes | We acknowledge the reviewer's point and recognise the need for clarification in the text. We would highlight that the objective of split-sampling is not validation/evaluation of the hydrological model. Rather, the aim is to test whether the selected parameter set is climatically and/or geographically transposable (Klemeš, 1986). We have adjusted the text to reflect the above and trust this provides the necessary clarification. |
| 9 | In the discussion (chap. 4.4) the covariance approach is suggested as a tool to be used in environmental flow assessment, climate change studies, water resources management, prediction in ungauged catchments, and model selection frameworks. I wonder if the approach is really a solution to many of the biggest challenges in hydrology? | Yes | This was not our intention. We have attempted to adjust the tone of the paragraph to better reflect the *potential* applications of the framework. |
| 10 | The results of this study are mostly compared against three other studies that model ER HIs (i.e., Pool et al., 2017; Vis et al., 2015, Shrestha et al., 2014). Could you maybe find more studies for comparison? There are quite a number of studies using hydrological signatures (not necessarily ecologically relevant) for model calibration or validation. Such studies could be helpful to discuss the covariance approach and the simulation results of this study. | Yes | The introduction and methods of the paper clearly set out a focus on ER HIs, therefore, we feel that a focus on hydroecological case studies is consistent with the objective of the paper. Additionally, previous reviews of the literature found that more generic studies focus on aspects of the flow regime that are not consistent with ER HIs or the facets of the flow regime. However, effort has been made to refer to some more recent studies where points of comparison may be possible. We trust this represents a suitable compromise. |
| 11 | P18 L20 - Study areas: Could you maybe provide some information on how exactly you selected the 5 study catchments? | Yes | Additions have been made to the indicated paragraph, with some minor restructuring to improve the overall clarity. |
| 12 | P18 L28 and Table 1: To complete the information in Table 1 I would add the information about the altitude, especially since you explicitly mention that altitude is different between the 5 study catchments. | Yes | Gauge altitude, catchment maximum altitude and an indicator of catchment steepness have been added. Catchment steepness is not available for the Tarland Burn as this is not part of the National Riverflow Archive. |
| 13 | I would also suggest to either provide the definition of BFI in the table caption or not to use the abbreviation. | Yes | This has been modified accordingly. |

| | | | |
|---|---|---|---|
| 14 | The * behind "principle land use" should probably also be added to the table caption. | Yes | The symbol was not related to any footnotes and has been removed |
| 15 | Finally, you could eventually add the river type to the table. | Yes | It is not possible to add the river type under WFD as each catchment is a sub-catchment of a much larger parent catchment and water management area. However, additional information has been added regarding how the WFD river types are defined. Building on this, additional information regarding these categories have been added to the table. The rivers were not classified using this data as it was felt this may be misleading given that they have not been formally classified under the WFD. |
| 16 | P19 L8: You mention that Perrin et al. (2008), Coron et al. (2016), and others have been using the GRJ model series for a wide range of applications. To increase the information content of this sentence, I recommend to explicitly state what GRJ has been used for (e.g. GR4 was used for climate change modelling, the prediction in ungauged basins, etc.) | Yes | Three example applications have been added, as well as an additional reference to support this (Rojas-Serna et al, 2006). |
| 17 | P19 L10-20: Parameter x3 is missing in the description of the model parameters. Given that the focus of the study is not the development of a new model I wonder if such a detailed description is needed. I suggest keeping the model description more generic focusing on the runoff processes represented by the model. | Yes | The text has been simplified to be more generic. Some detail has been retained for the interpretation of the model parameterisation. We apologise for the omission of X3, this has now been corrected. |
| 18 | P20 chpt. 2.4: The covariance approach is closely related to the GLUE methodology proposed by Beven and Binley (1992). Both approaches generate in a first step a large set of parameters, from which a subset is selected using limits of acceptability. Where do you see the differences? | NA | See detailed responses. |
| 19 | P20 L14-16: To me it is not clear from the context of the paragraph for which model (hydroecological model or hydrological model) the described data is used. | Yes | The use of the term model has been pre-fixed with hydrological/hydroecological in this paragraph as well as the preceding one. |
| 20 | Also, references to data sources are missing. | No | We respectfully disagree with the reviewer's comment. The paragraph directs the reader to Table 1 where the data sources are clearly indicated. |
| 21 | P21 Figure 1: The last column of stage 1 uses three boxes to show the model output. I would suggest to reduce it to two boxes. This could either be i) one box for covariance (observed and simulated) and one box for HI (observed and simulated) or ii) one box for simulations (covariance and HI) and one box for observation (covariance and HI). | Yes | We thank the reviewer for their suggestion. The figure has been simplified to (ii) |

| | | | |
|---|---|---|---|
| 22 | P22 L7-9: Could you shortly state why exactly you selected an e function and not a linear function? | Yes | Additional information added. |
| 23 | How did you decide that you want 20 parameter sets and not e.g. 30? | Yes | Additional information added. |
| 24 | P22 Fig. 2: The figure could be complemented by a legend to make it easier to understand. I would also suggest to use the grey area to highlight the plausible parameter space and not the parameter sets not used for the final simulations. | Yes | Figure has been relabelled, a legend has been added, and lines recoloured to allow for the change in background colour. |
| 25 | P24 L6-8: The range of parameter values is used as an indicator for consistency. However, the possible/plausible values of the model parameters vary in order of magnitudes. E.g. x4 that is an indicator for the event length (days) has different plausible values than x1 that is an indicator for the storage (mm). This leads to your result that those parameters having values in the order of 100 have a much lower consistency than those having values in the order of 1- 10. I would either normalize the parameter values or not use the range as in indicator for consistency. | Yes | Change made. |
| 26 | P 27 L9: NSE is a measure where the hydrological model simulations are evaluated against the simple model of predicting mean discharge for every day. | No | Thank you for the suggestion. In this case we have not followed the suggestion. We have amended Table 2 as per comment 34 and thus reduced the information content in this paragraph accordingly. |
| 27 | Section 3.2.5: In this section you compare the efficiencies across your study catchments and you don't present any new results. So I would move the whole section into the discussion chapter | Yes | Change made. |
| 28 | Section 4.1: This section is a summary of your results chapter. In the discussion, to me it would be more interesting to read about why you have certain results. You actually do that in section 4.3.1. I would therefore merge section 4.1. and 4.3.1. | Yes | We agree that the order of paragraphs sections here is not quite right. However, in order to be consistent with the aims set out in the introduction (regarding performance and consistency) it is necessary to address. With this and the reviewer's comments in mind, we have rearranged the sections to: 4.1, 4.3, then 4.2. We hope this improves the flow and achieves an effect consistent with the reviewer's request. |
| 29 | P31 L1: I don't think that your statement "… practical limit to the number of ER HIs" is correct. Because if one e.g. uses an optimization algorithm with a multi-criteria objective function, then each criteria can have a certain weight (equal weights or as in your case weights according to importance). So I don't think that the number of criteria is limited from a practical perspective | Yes | We agree, this is an error in the rewrite based on the original draft ("given the inherent limitations on the number of indices which may be considered at a given time"). This text has been adjusted so that it is clear that the limitation is that you do not know which indicators are important. |
| 30 | P18 L23: What does WFD mean? | Yes | Definition added. |

| 31 | P18 L23-25: I think the sentence "Under the modified covariance…." doesn't fit into the context of this paragraph and I would remove it. | Yes | This has been removed and the text restructured to improve the overall flow. |
|---|---|---|---|
| 32 | P20 L1: I would rename the chapter from "data" to e.g. "Selection of ecologically relevant hydrological indicators", because most information in this chapter is on the selection process of the ER HI. | Yes | The chapter has been renamed. Selection has been replaced by determination. Further amendement in the text to ensure consistency.
The title of the following section has been moved up to encompass the data which serves as input to the hydrological modelling. This may also serve to address the lack of clarity re which model type (hydroecological and hydrological) is being discussed. |
| 33 | P20 L1-10: You provide a description of the hydroecological modelling approach in the appendix. I would therefore shorten this paragraph to the minimum information needed to understand how you selected the ER HIs. Sentences such as "measure of the statistical weight…" or "Consequently, more conclusive statements may be made…" could be removed. However, I would complement the section with the information on how many ER HI were selected for each catchment, because this information is relevant for stage 3 in your modelling approach. | Yes | We thank the reviewer for their suggestion. The text has been reduced in a number of instances. However, we feel that the sentence, "Consequently, more conclusive statements may be made…" is necessary. One of the conclusions in Pool et al was a need to understand the importance of indicators prior to modelling. |
| 34 | P23 Table 3: I would extend the description of NSE: "…A measure of goodness of fit to the 1:1 line normalized by the variance". The detail on the normalization is important and you refer to in the discussion. | Yes | Change made. |
| 35 | P24 Table: Table 2 should be relabelled to table 4. | Yes | We apologise for this error - the field was updated by word automatically. This has now been addressed. |
| 36 | I also suggest to add parameter x6 for completeness. | No | We respectfully disagree with the reviewer's comment. The addition of parameter x6 would require the addition of a further column for each catchment, we feel that this would add unnecessary confusion.
Instead, we propose an addition to the caption to explain the omission of the parameter x6 from the table. |
| 37 | Figures 3, 4, and 5: Please explain what the subplots titles, such as D, F, T, and R mean. | Yes | We apologise for this omission; this has now been added to each caption. Further, to ensure consistency, the (real) Table 2 has been modified to also include these short-forms. |
| 38 | I made the general observation that many Figures are not readable in black and white. Can you think of a way to change that? | Yes | We apologise for this error. All figures should have been using the Viridis colour palettes to ensure the selection of print and colourblind friendly colours. This has now been corrected. |

**Reviewer 1, comment number 5**

*First, step 1 consists of sampling the parameter space and running simulations. So strictly speaking, simulations are run without validating the structure. To me a validation of the model structure without parameterization would e.g. be the check if the model structure represents the most important processes occurring in a catchment. For this no parameters are needed.*

*Second, stage 2 of the approach is the visualization of the simulations in the form of efficiency plots, and stage 3 is the selection of parameter sets using covariance and ER His. Stage 2 is therefore not a validation – validation is only done in step 3 at the moment where a "judgment" is made about the validity of a parameter set.*

**Response**

We would highlight that it is not validation of the parameter set we refer to, it is the validation, or invalidation, of the model itself. As highlighted by Vogel and Sankarasubramanian (2003), under a traditional split-sample approach, the model is calibrated, a parameter set is selected, and the model is validated for that parameter set. The covariance approach does not seek to validate a single parameter set, rather, the consideration of the range of the parameter sets is the validation. Consequently, it was possible to invalidate and reject the model GR6J.

However, irrespective of the above, a review of the literature indicates that there is no uniform agreement on a methodology for model validation, for example Vogel and Sankarasubramanian (2003) and Biondi et al. (2012). Indeed, Biondi et al. (2012) go on to identify three types of validation.

It is also apparent that there is a philosophical issue with the use of the term validation, with the term valid implying a degree of truthfulness that is misleading. Indeed, Oreskes and Belitz (2001) argue that l terms, such as evaluation or assessment, should be used in place of validation.

With the above, and the reviewer's comments, in mind, we have modified the paper to adopt a more neutral tone. We trust this addresses the concerns raised by the reviewer.

**Reviewer 1, comment number 7**

*I am still not so happy with the term covariance approach, because the selection of parameter sets is based on two covariance metrics and many more ER HIs. The approach is therefore named after one out of many metrics used.*

**Response**

We acknowledge the reviewer's point, but respectfully disagree for the following reasons.

1) First and foremost, it is necessary to term the methodology the covariance approach in order to ensure consistency with the original work by Vogel and Sankarasubramanian (2003):

"Our approach is to augment the current calibration/validation paradigm with a method which does not focus solely upon the traditional ''goodness of fit'' of the model predictions to the observations. Rather, our approach focuses on the ability of the hypothesized watershed model structure to represent the observed covariance structure of the input and output time series without ever calibrating the model. Since our approach focuses on the modeled covariance structure of the input and output series, we term our approach a covariance approach to model validation."

2) We never consider covariance in isolation; therefore, we would argue that it is not one of many metrics used. Covariance is always considered in conjunction with the other metrics as set out in the table below.

| Parameter set | $\rho(P, Q)$ | HI1 | HI2 | … | HIn |
|---|---|---|---|---|---|
| 1 | | | | | |
| 2 | | | | | |
| 3 | | | | | |
| 4 | | | | | |
| 5 | | | | | |

3) Additionally, a body of research which refers to the methodology as the modified covariance approach has been published (Visser et al., 2019a; Visser et al., 2019b). We would therefore argue that, to ensure consistency, the name of the approach should remain as is.

**Reviewer 1, comment number 18**

*P20 chpt. 2.4: The covariance approach is closely related to the GLUE methodology proposed by Beven and Binley (1992). Both approaches generate in a first step a large set of parameters, from which a subset is selected using limits of acceptability. Where do you see the differences?*

5 **Response**

Our understanding of the GLUE methodology as put forward in (Beven and Binley, 1992) is outlined below.

Parameter ranges and the sampling strategies are specified. From this, a large set of parameters are determined, and simulations run. In this, there is no difference between the two approaches.

*GLUE*

10 Thereafter, the methodologies have little in common. GLUE sees the determination of *likelihood* for each model realisation. This likelihood measure is determined by the nature of the prediction problem; for example, if the focus is on flood peaks then the measure should reflect the accuracy in replicating this aspect of the flow regime. All models where likelihood > 0 are used to weight the parameter set accordingly. The limits of acceptability in GLUE are generally determined a priori, and then adjusted accordingly (Beven, 2006).

15 *Covariance approach*

Under the covariance approach, the focus is on the covariance structure of the input and outputs of the model.

This approach focuses on joint variability whereas GLUE focuses on likelihood based on a performance measure. Performance measures are not considered under the covariance approach. Indeed, performance measures are limited in their ability to focus on the accuracy in replicating a range of hydrological indicators (i.e. one of the motivations for this study).

20 The covariance is related to each indicator and the limits of acceptability determined by statistical importance. GLUE does not allow for the consideration of a diverse range of indicators nor the consideration of the statistical importance of ER HIs.

The limits of acceptability may be specified apriori, as in Beven, or a number of parameter sets specified, as was done here.

The focus on covariance and the specific replication of ER HIs are the differentiating factors which set the covariance approach apart. We view the principle similarity in these two approaches to lie in the terminology associated with certain steps. It is also
25 worth noting that Vogel and Sankarasubramanian (2003) do not liken their approach to the already well-established GLUE methodology.

[revised manuscript text omitted]

---

## Author Response (AR3)

Dear Professor Pierre Gentine,

We would like to thank you and the reviewers for your reconsideration of the manuscript. We have made every effort to address the issues raised by the reviewers; for details see our responses below. We trust this will allow you to sign off on the paper happily.

Yours sincerely,

Annie Visser-Quinn (Corresponding author)

**Reviewer 1**

P1L12: fix "From this the plausible..."

P3L9: fix "focussed"

P3L10: add "but" or "while" prior to "the ability of the..."

P7L11: simplified exemplar --> simplified example

P14L21: The subsection title of "4.1.1" should be on a new line, and the preceding paragraph should not be in BOLD.

P16L21: subsection 4.2.2??

P17L5: The subsection title of "4.3" should be on a new line, and the preceding paragraph should not be in BOLD.

**Response:** We thank the reviewer for their useful editorial notes, the changes have been made as suggested. We have not changed the spelling of focussed as we have been consistent in the use of British-English throughout. We trust that the copyeditor will ensure the consistency in the style of English, whether it be British or American.

**Reviewer 2**

1) P11 L 24-25: The introduction now includes a part that defines the "traditional" approach. However, the definition is still relatively vague, mostly because there is a lack of examples. Is the Nash-Sutcliffe efficiency a traditional objective function? What about the Kling-Gupta efficiency? What about the flow duration curve? These questions can be easily clarified by adding the information (e.g. "…traditional objective functions, such as the NSE and KGE…"). Also, it would be interesting to know which objective function was used in the studies that are referenced right after the definition (Shresta et al., 2014; Vis et al, 2015). This would certainly help to define what a traditional objective function is.

**Response:** We have made a minor change to the introduction in an effort to better reflect our meaning of the traditional approach. The emphasis is on objective functions AND algorithms is the traditional approach, not the functions themselves.

2) P12 L10: You mention the selection of ER HIs to guide model parameterisation as one limitation you want to address in this study. The limits of acceptability concept of the covariance approach also needs a selection of ER HIs. Can you shortly comment on where the difference is?

**Response:** We would like to stress that the focus is not on the selection of ER HIs, but rather the identification or determination. The semantics are very important in the context of this work.

Under the proposed approach, the statistical importance is a vital part in the model parametrisation. As Pool et al. (2017) highlight, no other approach considers the relative importance of the indicators, this removes the current issue of subjectivity (see 4.2.2). We feel that this is sufficiently discussed throughout the paper.

3) P12 L11 and P28 L27-32: Is the recalibration of a model, such as GRJ, really a limitation in terms of computational power? I assume the bigger challenge is rather to find a set of parameters that manages to replicate multiple ER HIs. Having such a "common" set of parameters is hydrologically more meaningful than having a separate set of parameters for different ER HIs.

Please note that this statement was made with reference to Pool et al. (2017).

4) Table 1: Catchment steepness is missing for the Tarland Burn catchment. I assume that it could be calculated relatively easily using the catchment outline and a digital elevation model.

**Response:** The Tarland Burn is not an NRFA catchment, therefore the data availability is considerably less. The James Hutton Institute who provided the data could not provide a suitable digital elevation model for the area. To avoid confusion, we simply state in the Table caption that the data was not available.

5) This comment is on step 2 and 3 of the covariance approach. In Fig. 1 and the corresponding text you make a clear distinction between step 2 and step 3. I am not convinced that it is meaningful to separate the two steps. To my understanding you do the following: In step 2, you make a plot of the observed and n simulated covariances and each ER HI. If the observed moments lie inside the "cloud" of simulated moments, then the model structure is valid. In step 3, you take a subset of the "cloud" that is within the limits of acceptability. This gives you the valid parameter sets. You therefore use a stricter test to validate parameter sets than to validate the model structure.

My question is now: does that make sense?

An extreme example: imagine you have 10,000 parameter sets, a validated model structure, but only 1 parameter set is within the limits of acceptability. Would you say you have a valid model structure if only 1 parameter set actually manages to give you simulations of covariance and ER HI that are realistic enough to work with? Personally, I would doubt that my model structure is valid.

Based on your text and comments I understand that step 2 is exactly as proposed by Vogel and Sankrasubramanian and that step 3 is the extension. However, I think what you want to do is modifying their approach to 1) use it for multiple indicators, and 2) to weight indicators according to their importance. To do so, you can skip step 2 and directly apply step 3.

**Response:** We would argue that the evaluation of the model structure in this way is central to the approach. On page 6 we discussed that model parameterisation typically presupposes that the selected hydrological model is able to capture the underlying processes in the given catchment without evidence base. We consider model evaluation a separate step to emphasise the importance.

Additionally, we would like to highlight that the limits of acceptability are ultimately determined by the user.

6) P28 L23: As you mention, equifinality is reaching the same outcome by different means. This is mainly due to uncertainties in data and model structure/parameters and the limited type of data used to evaluate the model. Given the same model and data, I don't fully understand why the covariance approach reduces equifinality compared to a traditional approach.

5 **Response:** We would argue that we do not make this claim. On page 17 we are careful to state that the "uncertainty associated with equifinality is reduced". By this we mean that the epistemic uncertainty associated with accounting for equifinality, rather than the uncertainty of equifinality. To improve clarity, we have modified the text accordingly.

7) P30 L8: You mention that an improvement in consistency is reached by the covariance approach. Is this due to the covariance approach or due to the fact that all available ER HIs are used to select parameter sets, i.e. the ER HIs are part of

10 the calibration?

**Response:** We would like to highlight that on page 15 we note that no approach has been able to achieve this level of consistency prior (despite their use of ER HIs).

8) Finally, I fully agree that your approach is different from the GLUE approach proposed by Beven and Binley (1992). However, they have a lot of similarity and this is why I think it is fair to shortly comment on that somewhere in the

15 manuscript (two to three sentences are enough). I would like to add some thoughts to your previous answer to this topic: a) both, the GLUE and the covariance approach need a performance metric (you use the covariance and ER HIs; any objective function (also called likelihood function) can be used in GLUE), b) statistical importance could be considered in GLUE using the approach of fuzzy limits of acceptability (Beven, 2006), and c) a range of indicators can be considered in GLUE (e.g.Blazkova and Beven, 2009).

20 **Response:** We take the points of the reviewer on board. However, the focus of our paper is on improvements over the described traditional approach. Whilst GLUE is commonplace in hydrological modelling, it is not in the parameterisation of hydrological models for the assessment of hydrologic alteration. We therefore feel that the discussion of GLUE is outwith the scope of this paper. Additionally, we would again like to highlight that Vogel and Sankarasubramanian (2003), in their more general paper, did not liken their approach to the GLUE methodology.

[revised manuscript text omitted]
 | C1 | C2 | C3 | C4 | C5 | C6 | C7 | C8 | C9 | C10 | C11 | C12 | C13 | C14 | C15 | C16 | C17 | C18 | C19 | C20 | C21 | C22 | C23 | C24 | C25 |
|---|---|---|---|---|---|---|---|---|---|---|---|---|---|---|---|---|---|---|---|---|---|---|---|---|---|---|---|
| PlsFld | No. of pulses above a (baseline) flood threshold. | Count | | | | | | | | | | | | | | | | | | | | | 0.41 | 100 | 0 | 0 | 55.6 |
| PlsQ25w | No. of pulses above a Q*xx* (baseline) threshold. | Count | | | | | | 0.64 | 0 | 91.7 | 83.3 | 95.8 | | | | | | | | | | | | | | | |
| PlsQ25s | | Count | | | | | | 0.58 | 0 | 66.7 | 70.8 | 83.3 | | | | | | | | | | | | | | | |
| PlsQ50 | | | | | | | | | | | | | 0.04 | 0 | 100 | 100 | 100 | | | | | | | | | |
| PlsQ75 | No. of pulses below a Q*xx* (baseline) threshold. | Count | 0.69 | 0 | 0 | 0 | 0 | | | | | | 0.03 | 0 | 100 | 100 | 100 | | | | | | | | | |

*Timing*

[revised manuscript text omitted]